# Characterizing the oceanic ambient noise
# as recorded by the dense seismo-acoustic Kazakh network

Alexandr Smirnov[1,2], Marine De Carlo[3], Alexis Le Pichon[3], Nikolai M. Shapiro[4,5], Sergey Kulichkov[6]

[1]Institute of Geophysical Research, Almaty, 050020, Kazakhstan

[2]Institut de Physique du Globe de Paris, Sorbonne Paris Cité, F-75005 Paris, France

[3]CEA, DAM, DIF, F-91297 Arpajon, France

[4]Institut de Sciences de la Terre, Université Grenoble Alpes, CNRS (UMR5275), Grenoble, France.

[5]Schmidt Institute of Physics of the Earth, Russian Academy of Sciences, Moscow, Russia

[6]A.M. Obukhov Institute of Atmospheric Physics RAS, Moscow, 119017, Russia

*Correspondence to*: Alexandr Smirnov (smirnov@ipgp.fr)

**Abstract.** In this study, the dense seismo-acoustic network of the Institute of Geophysical Research (IGR), National Nuclear Centre of the Republic of Kazakhstan is used to characterize the global ocean ambient noise. As the monitoring facilities are co-located, this allows for a joint seismo-acoustic analysis of oceanic ambient noise. Infrasonic and seismic data are processed using a correlation-based method to characterize the temporal variability of microbarom and microseism signals from 2014 to 2017. The measurements are compared with microbarom and microseism source model output that are distributed by the French Research Institute for Exploitation of the Sea (IFREMER). The microbarom attenuation is calculated using a semiempirical propagation law in a range-independent atmosphere. The attenuation of microseisms is calculated taking into account seismic attenuation and bathymetry effect. Comparisons between the observed and predicted infrasonic and seismic signals confirm a common source mechanism for both microbaroms and microseisms. Multi-year and intra-seasonal parameter variations are analysed, revealing the strong influence of long-range atmospheric propagation on microbarom predictions. In winter, dominating sources of microbaroms are located in the North Atlantic and in the North Pacific during Sudden Stratospheric Warming events, while signals observed in summer could originate from sources located in the southern hemisphere, however additional analyses are required to consolidate this hypothesis. These results reveal the strengths and weaknesses of seismic and acoustic methods and lead to the conclusion that a fusion of two techniques brought the investigation to a new level of findings. Summarized findings are also perspective for a better description of the source (localization, intensity, spectral distribution) and bonding mechanisms of the ocean/atmosphere/land interfaces.

## Introduction

Since the original research of Bertelli (1872), many investigations have confirmed a close connection between microseisms and disturbed ocean weather conditions (Longuet-Higgins, 1950). The primary microseism peak (around 0.07 Hz) is generated when ocean waves reach shallow water near the coast and interact with the sloping seafloor (Hasselmann, 1963). The secondary peak of microseisms (between 0.1 and 0.2 Hz) is generated by the interaction of ocean waves of similar frequencies travelling

in opposite directions (Longuet-Higgins, 1950). Longuet-Higgins' theory explains how counter propagating ocean waves can generate propagating acoustic waves and create secondary microseisms by exciting the sea floor. Hasselmann (1963, 1966) generalized Longuet-Higgins' theory to random waves by investigating non-linear forcing of acoustic waves.

Microseism modelling was introduced by Kedar et al. (2008). The good correlation between the observed microseism amplitudes and their predicted values was shown (Shapiro, 2005; Shapiro and Campillo, 2004; Stehly et al., 2006; Stutzmann et al., 2012; Weaver, 2005). The different patterns between microseismic body and surface waves, resulting from the amplification of ocean wave-induced pressure perturbation and seismic attenuation have been studied with implications for seismic imaging and climate studies (Obrebski et al., 2013). Coastal reflections also play an important role in the generation of

microseisms, but modelling ocean wave reflections off the coast still remains a major source of model uncertainty (Ardhuin et al., 2013a). Ardhuin and Herbers (2013b) developed a numerical model based on Longuet-Higgins-Hasselmann's theory for the generation of Rayleigh waves, by considering an equivalent pressure source at the undisturbed ocean surface.

    Inaudible low-frequency sound, known as infrasound waves, propagate through the atmosphere for distances of thousands of kilometres without substantial loss of energy. Below 1 Hz, infrasound has been observed since the early nineteenth century at

different locations distributed around the globe. Gutenberg (1953) first pointed out the relation between microseisms, meteorological conditions, ocean waves, and microbaroms. Donn and Naini (1973) suggested a common source mechanism of microbaroms and microseisms from the same ocean storms demonstrating that the only mechanism capable of transmitting energy into both the atmosphere and the sea bottom is associated with surface wave propagation.

    There is a significant difference between microseisms and microbaroms. While propagation paths for microseisms can be either

along the earth's surface as Rayleigh waves, or through the Earth as body waves (Gerstoft et al., 2008), microbarom observations are typically along propagation paths that have undergone multiple bounces on the Earth surface. As for microseisms, microbaroms are not impulsive signals but quasi-monochromatic sequences of permanent waves (Olson and Szuberla, 2005); therefore, it is not possible to detect their onset and identify their propagation paths. However, these signals are well detected using standard array processing techniques, such as beamforming methods (Capon, 1972; Haubrich and McCamy, 1969;

ToksöZ and Lacoss, 1968). Several studies demonstrated the efficiency of beamforming approaches (e.g. Evers and Haak, 2001), or correlation-based methods (e.g. Garcès, 2004; Landès et al., 2012), to detect and characterize microbarom signals globally. Posmentier (1967) started developing a theory of microbaroms based on the Longuet-Higgins' theory. A microbarom source model was first developed by Brekhovskikh (1973), later extended by Waxler and Gilbert (2006), 60 Waxler (2007), and more recently by de Carlo (2020).

Losses along the propagation path control the ability to observe microbaroms. Thus, in order to accurately assess the microbarom source intensity, it is necessary to take into account a realistic description of the middle atmosphere. Several studies have been conducted to characterize the ambient infrasound noise. Smets et al. (2014) compared microbarom observations with predicted values to study the life cycle of Sudden Stratospheric Warming (SSW). Landès et al. (2014) compared the modelled

source region with microbarom observations at operational stations of the International Monitoring System (IMS). A first-order agreement between the observed and modelled trends of microbarom back-azimuth was shown. Le Pichon et al. (2015) compared observations and modelling over a 7-month period to assess middle atmospheric wind and temperature models distributed by European Centre for Medium-Range Weather Forecasts (ECMWF). It was shown that infrasound measurements can provide additional integrated information about the structure of the stratosphere where data coverage is sparse. More recently, Hupe at al. (2018) showed a first order agreement between the modelled and observed microbarom back-azimuth and amplitude in the Northern Atlantic.

In this paper, we develop a synergetic approach to better constrain microbarom source regions and evaluate propagation effects. To this end, we apply the method developed by Hupe et al. (2018) to the dense Kazakhstani seismo-acoustic network. The considered network is operated by the Institute of Geophysical Research (IGR) of the National Nuclear Centre of the Republic of Kazakhstan. It includes both seismic and infrasound arrays. Since the pioneering work of Donn and Naini (1973), to our knowledge, this study is the first multi-year comparisons between observed and modelled ambient noise at co-located seismoacoustic arrays. In the first part, we have presented the observation network and the methods used. In the second part, the processing and modelling results of microseism and microbarom signals recorded by the IGR seismo-acoustic network from 2014 to 2017 are shown. In the last part, comparisons between the observed and modelled microbaroms and microseism are discussed.

## 1    Observation network and methods

### 1.1    Observation network

#### 1.1.1    Infrasound array network

The Kazakhstani seismo-acoustic network (KNDC, 2019) contains five seismic and three infrasound arrays (Figure 1). The signal correlation in such a dense network is significantly higher compared to sparser networks like the IMS. The infrasound network consists of the IMS station IS31 located in North-West Kazakhstan (2.1 km aperture, 8 elements) and two national arrays of 1 km aperture: KURIS (4 elements) near Kurchatov and MKIAR (9 elements) near the village of Makanchi (Belyashov et al., 2013). KURIS and MKIAR have been operating since 2010 and 2016, respectively. Microbarometers MB2000 and MB2005 are used at IS31 and KURIS, and Chaparral Physics Model 25 microbarometers are installed at MKIAR. All arrays are equipped with a 24-bit digitizer with a sampling frequency of 20 Hz at IS31 and KURIS, and 40 Hz at MKIAR. Data logger parameters are listed in Table A1 (Appendix A). All stations are equipped with 96 port wind noise reducing system with pipe rosettes, except L1, L2, L3, and L4 elements at IS31 which are connected to 144 inlet ports (Marty, 2019). The frequency response of the microbarometers are shown in Figure A1 (a,b). By associating infrasound observables over the network, both natural and anthropogenic infrasound sources can be detected and characterized (Smirnov, 2015; Smirnov et al., 2011 and 2018).

 **1.1.2    Seismic array network**

The seismic network consists of Kurchatov Cross array and MKAR that are part of the IMS network, as well as ABKAR and KKAR arrays which are part of the Air Force Technical Applications Centre (AFTAC, USA) network (Figure 1 and Table 1). Kurchatov cross array consists of 20 Guralp CMG-3V sensors with an aperture of ~22.5 km (Figure 1). ABKAR, BVAR, KKAR and MKAR arrays consist of nine elements with an aperture of ~5 km. These arrays are equipped with Geotech Instruments GS21 short period vertical sensors with a flat response for frequencies above 1 Hz. The frequency response of the sensors at MKAR, ABKAR, and KKAR is not flat in the 0.1-0.3 Hz band; however, as the response information is given, one can correct for the drop in amplitude; the phase shift difference between instruments part of the same array is assumed negligible. Figure A1 (c,d) shows the frequency response of GS-21 and CMG-3V sensors between 0.1 and 0.4 Hz. All arrays are equipped with 24-bit digitizers, sampling data at 40 Hz. Surface waves from the ocean storms are well recorded by broadband seismometers. Body waves are also registered by GS21 short period sensors. Although, in the frequency band of interest the signal attenuation is about 30 dB, all stations detect microseisms due to their large amplitude above the background noise.

A peculiarity of the network is that infrasound and seismic arrays are collocated at two sites (KURIS and Kurchatov Cross; MKIAR and MKAR), or installed relatively close to each other (IS31 and ABKAR are 220 km apart, Figure 1). Figure B1 shows typical power spectral density (PSD) of the ambient noise at infrasound and seismic arrays, and at collocated Kurchatov cross seismic and KURIS infrasound arrays. PSD calculation was carried out using one-hour time window during calm periods in October, December and July. The microbarom peak is more pronounced in October and December. In summer, this peak is only visible at IS31. As opposed to the infrasound noise, the seismic noise spectra exhibit the microseismic peak in both seasons with an overall noise level in October approximately 10 dB higher than in July.

**1.2 Processing method**

Microseisms are detected using the Progressive Multichannel Correlation Method (PMCC) (Cansi, 1995; Cansi and Klinger, 1997; Smirnov et al., 2011) in 10 linearly spaced frequency bands between 0.05 and 0.4 Hz. A fixed time window length of 200 s is used for each band. For the infrasound processing, the frequency band is broadened to 0.01-4 Hz using fifteen logarithmically scaled sub-bands, and time window length varying from 30 s to 200 s (Matoza et al., 2013). Such setting allows computationally efficient broadband processing and accurate estimates of frequency-dependent wave parameters useful for source separation and characterization. In the microbarom frequency range covering 0.1-0.6 Hz interval, wave parameters can be detailed in 6 different frequency bands (Ceranna et al., 2019).

It is important to take into account uncertainties in azimuth and apparent velocity estimations identified in microbarom studies. The uncertainties of the estimated wave parameters of microseisms can be large due to the relatively small aperture of the arrays. Uncertainties in wave parameter estimates are calculated considering the array geometry of the above mentioned

infrasound and seismic arrays, assuming perfectly coherent signals and time delay errors bounded by twice the sampling period (Szuberla and Olson, 2004) (Table 1). For the infrasound arrays, the horizontal speed is set to 0.34 km/s. For the seismic arrays, a value of 3 km/s typical Rayleigh wave speed is chosen. The uncertainties for the seismic arrays are significantly higher for the body waves due to higher velocities. It should be noted that these errors are optimistic as the estimation do not take into account site and time dependent signal-to-noise ratio.

### 1.3 Source modelling

The used microseism source model (IFREMER, 2018), referred to as 'p21', is calculated from the wave-action WaveWatch III model (WW3) developed by the National Oceanic and Atmospheric Administration (NOAA). While the bathymetry strongly affects the source intensity in microseism modelling (Ardhuin et al., 2011; Ardhuin and Herbers, 2013a; Kedar et al., 2008), a recently modelling study by De Carlo (2020) suggests that bathymetry has negligible impact on microbarom source strength in contrast to predictions from the model by Waxler (2007). In this study, the source term for microseisms ('p2l') which does not include coupling with the bathymetry is taken as a proxy to model microbaroms. While microseisms propagate through the static structure of the solid Earth, long-range microbarom propagation is controlled by the strong spatio-temporal variability of the temperature and wind structure of the atmosphere. Therefore, the geometrical spreading and seismic attenuation are the main effects to account for microseism modelling (e.g. Kanamori and Given, 1981; Stutzmann et al., 2012), while the dynamical properties of the middle atmosphere should be taken into account for microbarom modelling.

#### 1.3.1 Microbarom source modelling

As previously stated, both microseisms and microbaroms originate from second order non-linear wave interactions. Their source term can be written as a function of the second-order equivalent surface pressure $F_p(f_2 = 2f)$ (Hasselmann, 1963, Ardhuin et al. 2011):

$$F_p(f_2 = 2f) = \frac{1}{2} \rho_w^2 \, g \, f_2 \, H(f) \qquad (1)$$

where $\rho_w$ is the water density, $g$ is the gravitational acceleration, $f_2$ is the microseisms and microbarom frequency. The Hasselmann integral $H(f) = \int_0^{2\pi} E(f, \theta) E(f, \theta + \pi) d\theta$ (Hasselmann, 1963) represents the amount of opposite propagative wave interactions, with $E(f, \theta)$ the directional spectrum of waves. The IFREMER's distribution of the wave action model WAVEWATCH III® (WW3 Development Group, 2016; ftp://ftp.ifremer.fr/ifremer/ww3/HINDCAST/SISMO) includes the calculation of $F_p(f_2 = 2f)$ with a 0.5°x0.5° spatial resolution and 3 h temporal resolution.

Longuet-Higgins (1950) showed that these pressure fluctuations in the water do not attenuated with depth but are transmitted to the ocean bottom as acoustic waves. Depending on the ratio between the wavelength of the acoustic waves and the ocean depth, resonance effects can occur leading to a modulation of the pressure fluctuations at the sea floor (Stutzmann et al., 2012).

Therefore, microseisms are strongly affected by the bathymetry (Ardhuin et al., 2011; Ardhuin and Herbers, 2013a; Kedar et al., 2008). The corresponding seismic source power spectral density at the ocean bottom is (Longuet-Higgins, 1950; Eq. 184):

$$S_{DF}(f_s = f_2) = \frac{2\pi f_s}{\rho_s^2 \beta^5} [\sum_{m=1}^{m=N} c_m^2] F_p(f_2 = 2f) \qquad (2)$$

where $S_{DF}$ is in m/Hz, $\rho_s$ and $\beta$ are respectively the density and S-wave velocity in the crust, and coefficients $c_m$ correspond to the compressible ocean amplification factor. $c_m$ is non-dimensional number varying between 0 and 1 as a function of the ratio $2\pi f 2h/\beta$, where $h$ is the water depth. In this study, the crustal density $\rho_s = 2600$ kg m$^{-3}$ and the S-wave velocity $\beta = 2800$ m/s. The microbarom source term developed by De Carlo (2020) is essentially a scaled version of the second-order equivalent surface pressure $F_p(f_2 = 2f)$, which serves as proxy of microbarom source term.

### 1.3.2    Microbaroms propagation

For the propagation modelling, we use a semi-empirical frequency dependent attenuation relation derived from massive parabolic equation simulations (Le Pichon et al., 2012). Atmospheric specifications are extracted at the station from the high resolution forecast (HRES) that is part of ECMWF's Integrated Forecast System (IFS) cycle 38r2 (http://www.ecmwf.int) and are assumed constant along the propagation path. This approach, already used by De Carlo et al. (2018) and Hupe et al. (2018) to model microbaroms generated in the northern hemisphere, can predict the observed back-azimuths with an error less than ~10°. The correlation coefficient between the observed and predicted seasonal patterns is calculated following metrics elaborated by Landès et al. (2014). The correlation is evaluated for the back-azimuths and amplitudes.Two different metrics are derived: (i) $S_{corr\_Az}$ which defines the correlation between the observed ($N_{obs}$) and predicted ($N_{pred}$) marginal detection number in the direction $\theta_{Amax}$ versus time (t):

$$S_{corr\_Az} = C_{corr} [N_{obs} (\theta_{Amax}, t), N_{pred} (\theta_{Amax}, t)] \qquad (3)$$

and (ii) $S_{corr\_Amp}$ which defines the correlation between the predicted and observed amplitude $A_{max}$:

$$S_{corr\_Amp} = C_{corr} [N_{obs} (A_{max}, t), N_{pred} (A_{max}, t)] \qquad (4)$$

## 2 Results

### 2.1 Processing results

Signals from the ocean storms are extracted from detections at all IGR infrasound and seismic arrays, and filtered between 0.1 and 0.4 Hz. Diagrams in this section show the back-azimuths of the signals as a function on time. Distributions of the maximum amplitudes are included as well. The amplitude maxima are averaged over 6-hour time-window for the entire period of 20142017.

#### 2.1.1 Microbaroms

Figure 2 shows the temporal variation of the dominant microbarom signals at IS31, KURIS and MKIAR. The graphs show pronounced seasonal variations for both back-azimuths and amplitudes. The largest amplitudes at IS31 are observed during the winter months with a dominant period ranging from 3.5 to 5.5 s (Figure C1), when signals with back-azimuths of 320±20° prevail (Figure 2, a-b). Few detections with back-azimuths of 35±15° are also detected. In winter, microbarom amplitudes range from ~0.005 to ~0.5 Pa, the largest values being observed in winter. During summer months, signals with back-azimuths of 210±50° dominate with a period ranging from 4 to 6.5 s and lower amplitude (~0.01 Pa), suggesting waves propagating over longer epicentral distances. Figure 2 (e-h) shows the observations at KURIS. The back-azimuths measured at this station are similar to those recorded at IS31, with slightly higher values in winter (325±15°) and two clusters in summer at 230±30° and 120±30°. In summer, back-azimuths of 210±50° also dominate at IS31, KURIS and MKIAR. MKIAR started recording microbaroms in August 2016 with cyclical seasonal variations (Figure 2, i-e).

#### 2.1.2 Microseisms

Figure 3 (a-d) shows the detection results at ABKAR. In addition to the observations, the diagrams represent the simulated microseism parameters. The largest amplitudes are observed in winter where detections at 340±20° prevail. In summer, signals at 290±20° dominate. The amplitudes range from ~250 to ~10000 nm/s. Figure 3 (e-h) shows the results at KKAR. Two clusters of detections at 330±20° and 5±5° are observed in winter, and at 160±20° and 190±15° in summer. The seasonal amplitude variation is ~250 to ~9000 nm/s. Figure 3(i-l) shows the results at Kurchatov Cross. In winter, back-azimuths of microseisms are 300±20°. A small amount of detections at 50±50° is observed in summer. The amplitude ranges from 250 to 9000 nm/s, reaching their maximum values in winter. Figure 3 (m-p) shows results at MKAR. Two clusters at 310±20° and 5±5° are observed in winter, and at 130±10° and 180±10° in summer. The seasonal amplitude variation is ~250 to ~3000 nm/s. The seasonal trend of the microseism amplitudes recorded at all seismic stations is similar, with a maximum observed in winter. At Kurchatov Cross, the small amount of detections in summer could be explained by higher noise level or a loss of signal coherency at this site. The graphs clearly show that the amplitudes vary synchronously even at smaller time scale (Figure 4). As expected, the maximum amplitudes decrease with increasing distance from the stations to the North Atlantic region (about 10000, 9000, 9000 and 5000 nm/s for ABKAR, KKAR, Kurchatov Cross and MKAR, respectively).

## 2.2    Modelling results

The back-azimuths and amplitudes have been predicted at IS31, KURIS, and MKIAR. The distances to the source regions differ essentially from summer to winter. For example, simulations predict three source regions at IS31 in winter. Distances to the two regions in the North Atlantic are around 3500 km and 7000 km, and about 7000 km to the North Pacific. In summer, one source region is located in the Pacific Ocean and two other sources at Southern high latitudes at distances of ~12000 km and ~18000 km. However, the calculation of attenuation using a range-independent atmospheric model would inevitably lead to great mistakes in such situation. Figure 2 (a-l) compares the observed and predicted arrivals at these stations. In winter, a good agreement is found: IS31 records microbaroms with back-azimuths of 320±20° within the predicted range (Figure 2, a-c). A good agreement is also observed at KURIS (Figure 2, y-g) and MKIAR (Figure 2, i-k).

In summer, the agreement in azimuths remains satisfactory at all stations within a range of ±30°. IS31 records microbaroms within 210±50° with a slight shift compared with the predicted system (185±50°). At KURIS, the observed systems 230±30° and 130±30° are different compared with the predicted ones (±10° and 160±10°). At MKIAR, during the summer months, microbaroms are predicted with larger discrepancies (±70°). As the used source model was developed for microseisms (Ardhuin et al., 2011), an empirical scaling factor (F = 1:2600) has been applied to account for wave coupling effect in the atmosphere, thus allowing qualitative comparisons between the observed and predicted temporal variations of the microbarom amplitudes. Overall, at all stations, there is good agreement between the predicted and observed amplitudes during the winter months (Figure 2 d, h, l), but in summer, the predicted amplitudes are overestimated (Table 2). A first reason is that PMCC cannot detect multiple sources in the same frequency band. A second reason is the limitation of the propagation modelling which considers range independent atmosphere. It can be noted that the propagation anomaly predicted during of the SSW on January-February 2017 is not observed. Wind noise variations at the station, not considered in the simulations, could explain part of these discrepancies.

To summarize, both amplitudes and azimuths of the microbaroms are well predicted in winter as opposed to summer months. Microseism predictions show dominant source regions south of the arrays that are not observed. Quantitative estimations of the prediction quality ($S_{corr}$ calculated according to Eqs. 3 and 4) are summarized in Table 2.

## 3    Discussion

Where previous studies analysed microbarom signals at a single station (Hupe et al., 2018), further investigations are here conducted by considering a multi-year dataset of continuous records collected by the IGR network. Regional features of both microbaroms and microseisms are highlighted. Figure D1 (a-c) in Appendix D shows the azimuthal distribution of infrasound detections having maximum amplitudes. Figure D2 (a-d) shows similar histograms for seismic stations. One can distinguish seasonal trends for both infrasonic and seismic observations. In winter, microbaroms and microseisms are detected from the northern and northwestern directions. In summer, southern, southwestern and southeastern directions dominate; signals from

northwestern direction are also recorded at ABKAR, KKAR, and MKAR. Azimuths differ from one station to another depending on the strongest microbarom and microseism source regions relative to the station locations. Observations and simulations show large temporal variations in the dominating microbarom source regions explained by the seasonal reversals of the prevailing stratospheric winds, which in turn, cause the migration of storm activity area to the winter hemisphere. The histograms of the azimuthal distribution of microbaroms (Figure D1) clearly show the dominating direction of arrivals in winter with prevailing directions ranging from 270 to 350°. The predicted azimuths are in good agreement with the observed ones as shown by Figure 2 (c,g,k), Figure D1 and Table 2. In winter, microseism observations exhibit a similar pattern with a larger spreading (250-360°), and an additional peak (0-20°) at KKAR and MKAR (Figure D1, d-f). These peaks are explained by North Pacific microseism source regions.

In winter, microseisms exhibit similar trends with some differences as shown by Figure 3 (c,g,k,o). The dominant directions are comparable with a larger spreading: from 250° to 360° and from 0° to 20°. At KKAR and MKAR, two peaks are noted in the histograms, with a second peak at 0-20°. These peaks are explained by North Pacific microseisms. In summer, microbaroms are predicted mainly from the southern direction (180-200°). Such a peak is observed only at IS31 and MKIAR (Figure D1, c), although there is a large spreading in the predictions (45-225°). The closest peak observed at KURIS and MKIAR is shifted northwards by ~50°. The dominant back-azimuths are close to 90°. In winter, signals from ocean storms in the North Atlantic dominate at all stations. This is supported by microbarom and microseism simulations. Microbarom sources recorded by the Kazakh network in summer are not fully characterized. The cross-bearing location considering detections at IS31, KURIS, and MKIAR yields a hotspot located southwest of South America (Figure C2). Since the localization does not include crosswind effect, the true location may differ significantly from the preliminary estimation. Furthermore, the fact that a signal should pass a considerable portion of the way upwind, would prejudice the likelihood of its registration. However, this preliminary location is consistent with the relatively low amplitude values and larger periods in summer than in winter (Figure C1). Additional studies using more realistic propagation modelling are required to confirm this hypothesis. In this study, the method used to predict the attenuation assumes a range independent atmosphere along the propagation paths. Such an approach cannot be applied to situations involving long propagation ranges where significant along-path variability of wind and temperature profiles may occur (especially when sources and network are located in different hemispheres). Using historical IGR datasets, the spatiotemporal variability of microbarom signals due to changes in the source location and the structure of the atmospheric waveguides can be studied. There is a clear seasonal trend in both directions and amplitudes of microbaroms and microseisms (Figure 2). Moreover, microseism amplitudes synchronously vary at all stations (Figure 4). A good agreement between observations and simulations is found for the azimuths. The bathymetry effect plays an important role when calculating the microseism source intensity. As already shown by Evers and Siegmund (2009) and Smets and Evers (Smets and Evers, 2014), SSW events can be inferred from the observed spatio-temporal variations of microbarom parameters. Such observations are noted at IS31 where microbaroms in early and late February 2017 are shifted to easterly directions (~40°), which is consistent with the simulated source regions in the Northern Pacific (Figure 2 a, c). As noted at IS31, KURIS also recorded signals with

back-azimuths of ~40° in late January 2017 (Figure 2 e, g). Similarly, signals from ~100° were also recorded during the 2017 SSW event at MKIAR. However, the observed back-azimuths differ from the predicted ones (~60°). It is likely that this station recorded signals from other regions over the Pacific Ocean, which are not described by the used ocean wave model. These findings are consistent with comparisons between the observed and modelled microbaroms carried out by Landès et al. (2014) at IS31. This study shows that modelling well describes microbarom sources in the North Atlantic in winter, while signals in summer are poorly explained.

Comparing microbaroms and microseisms at collocated sites highlight similar features. Figure 5 (a-d) presents the observed back-azimuths and signal amplitudes from 1 January 2014 to 31 December 2017 at ABKAR and IS31, located 230 km apart. Figure 5 (e-h) shows the detection results for the collocated Kurchatov Cross and KURIS arrays. The comparison of the bulletins in Figure 5 shows similar seasonal patterns:

- North Atlantic microseisms and microbaroms prevail in winter. Back-azimuths of 300-360° are clearly visible in Figure 5 (a,b,e,g).

- Amplitudes of North Atlantic microbaroms and microseisms observed in winter exceed those observed in summer, as shown in Figure 5 (b,d,f,h).

Specific features are identified:

- Arrays record North Atlantic microseisms more steadily than microbaroms from that region (Figure 5).

- The range of back-azimuths for North Atlantic microseisms is larger than the ones of microbaroms at ABKAR and MKAR as shown by Figure 5 (a,b,e,g). In winter, at ABKAR, signals with back-azimuth of ~310° are predicted, while the observed signals dominate at ~340°. In summer, the signals predicted around ~180° are not observed (Figure 3 (a)). Such deviations in surface wave back-azimuths were earlier identified during teleseismic events observation at Alp Array (Kolinsky, 2019). To substantiate this hypothesis, Source Specific Static Corrections (SSSC) are required. However, the SSSC evaluation would require long-term instrumental observations, which is out of the scope of the present studies.

- In summer, no correlation is found in the prevailing directions of microseism and microbarom arrivals at collocated arrays.

This study aims at characterizing the oceanic ambient noise using infrasound and seismic methods. The results show that exploiting the synergy between seismic and infrasound ambient noise observations is valuable to: (i) better constrain the source strength using seismic records as microseisms propagate through the static structure of the Earth, while microbaroms travel through a highly variable atmosphere both in space in time, (ii) improve the detectability of ocean-wave interaction and location accuracy as microbarom wave parameters are less affected by heterogeneities in the propagation medium, and (iii) improve the physical description of seismo-acoustic energy partitioning at the ocean-atmosphere interface. While dominant features of

microseisms and microbaroms are successfully recovered, some limitations of the proposed approach are identified. One limitation is the inability of the PMCC method to detect signals from several sources overlapping in the same frequency band. Another methodological shortcoming is the range-independent atmosphere considered for propagation simulations. Such an approach cannot be applied to situations involving long propagation ranges where significant along-path variability of wind and temperature profiles may occur; especially when sources and network are located in different hemispheres. Additional studies are also required to further evaluate whether the bathymetry effect could explain discrepancies between the observed microbarom and microseism signals (Longuet-Higgins, 1950; Stutzmann et al., 2012, De Carlo, 2020).

**Conclusions**

The IGR seismo-acoustic network is much denser than the global IMS infrasound network. Analysing multi-year archives of continuous recordings provides a detailed picture of the spatial and temporal variability of the seismic and infrasound ambient noise originating from two hemispheres. In winter, the most intense oceanic storms are modelled in the Northern Atlantic and their signature prevails on infrasound and seismic records. During minor SSWs, bi-directional conditions may occur which may have strong impacts on the retrieved microbarom signals (Assink et al., 2014). Simulated and observed microbarom parameters are consistent, as shown by moderate correlation coefficients. In summer, the location of microbarom signals using detections at IS31, KURIS, and MKIAR is found southwest of South America, at a distance larger than 15000 km, near the peri-Antarctic belt where strong ocean storms circulate. This location is consistent with the relatively low amplitude and frequency of the recorded signals.

Further numerical investigations are needed to define the most suitable detection parameters in terms of missed events and false alarm rate, and estimate wave parameter uncertainties accounting for the response functions at all arrays. In this study, the discrepancies between observations and predictions motivate the use of high-resolution detection methods to identify multiple propagation paths from which microbarom energy can reach the array (e.g., Assink et al., 2014). Exploring the capability of high-resolution detection processing techniques to extract multidirectional overlapping coherent energy would be valuable to provide a more realistic picture of the recorded ocean ambient noise (e.g., den Ouden et al., 2020).

For such long propagation ranges, more realistic numerical simulations could reduce the differences between the observed and modelled amplitude; additional studies are thus required to explore time- and range-dependent full-wave propagation techniques while still maintaining computational efficiency (e.g., Waxler and Assink, 2019). Finally, including additional data from other seismo-acoustic networks worldwide would help constraining microbarom source location, validating long-range propagation modelling, and better characterize station-specific ambient noise signatures, which is important for a successful verification of the CTBT using the IMS.

**Code/Data availability**

Atmospheric wind and temperature profiles are derived from operational high-resolution atmospheric model analysis, defined by the Integrated Forecast System of the ECMWF, available at https://www.ecmwf.int/ (last access: 2 September 2019; ECMWF, 2018). Seismic and infrasound waveforms from the IMS network (https://www.ctbto.org/, last access: 2 September 2019) used in this study are available to the authors, being members of National Data Centres for the CTBTO. Data of the Kazakhstani national seismic and infrasound arrays are available under request to the Institute of Geophysical Researches, National Nuclear Centre of Kazakhstan. Microseism and microbarom detections of the seismo-acoustic Kazakh network and microbarom simulations are available at the ISC repository (Smirnov et al., 2020).

**Author contribution**

N. Shapiro and A. Le Pichon suggested main outlines of the paper. A. Smirnov and A. Le Pichon prepared historical dataset for processing. M. De Carlo and A. Le Pichon developed the microbarom source model. A. Smirnov performed microbarom and microseism detections and propagation simulations. A. Smirnov prepared the manuscript with contributions from all coauthors. A. Le Pichon, M. De Carlo and S. Kulichkov made critical reviews and comments to improve the manuscript.

**Competing interests**

The authors declare that they have no conflict of interest.

**Acknowledgements**

This research has been supported by the Commissariat à l'Energie Atomique (CEA, France). The work of Nikolai M. Shapiro has been supported by the European Research Council (ERC) under the European Union Horizon 2020 Research and Innovation Programme (grant agreement 787399-SEISMAZE), the Russian Ministry of Education and Science (grant no14.W03.31.0033) and Russian Foundation for Basic Research (project no. 18-05-00576). Authors also thank Anna Smirnova for support in the manuscript preparation, and Jelle Assink whose comments and suggestions helped improve and clarify the manuscript. The authors are also thankful to Inna Sokolova and Pavel Martysevich for valuable advices on the instrumentation part and Sven Peter Näsholm and Ekaterina Vorobeva for microbarom model scaling. Massive numerical computations were performed on the S-CAPAD platform of IPGP in France.

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

**Figures and Tables**

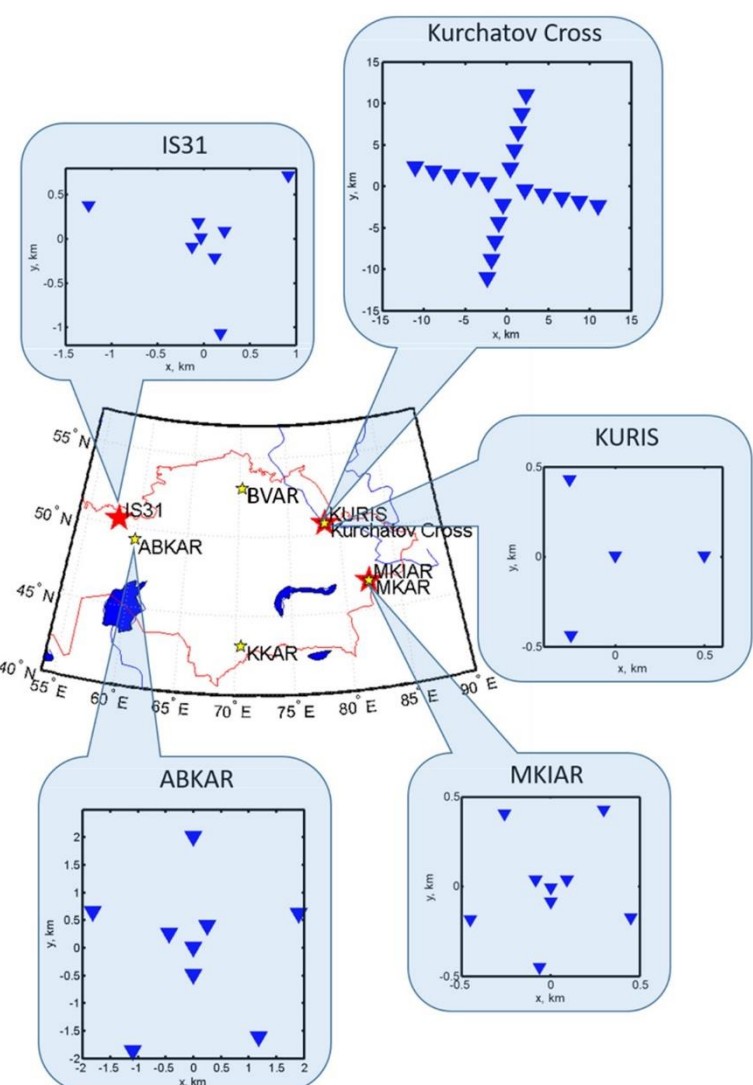

**Figure 1. IGR monitoring network. Yellow and red stars are seismic and infrasound arrays, respectively. Seismic and infrasound arrays are collocated at Kurchatov (Kurchatov Cross/KURIS) and Makanchi (MKAR/MKIAR)). IS31 infrasound and ABKAR seismic arrays are located ~200 km apart.** The **inset graphs show the array configurations. The configurations for KKAR and MKAR seismic arrays are not shown as they are similar to ABKAR's one.**

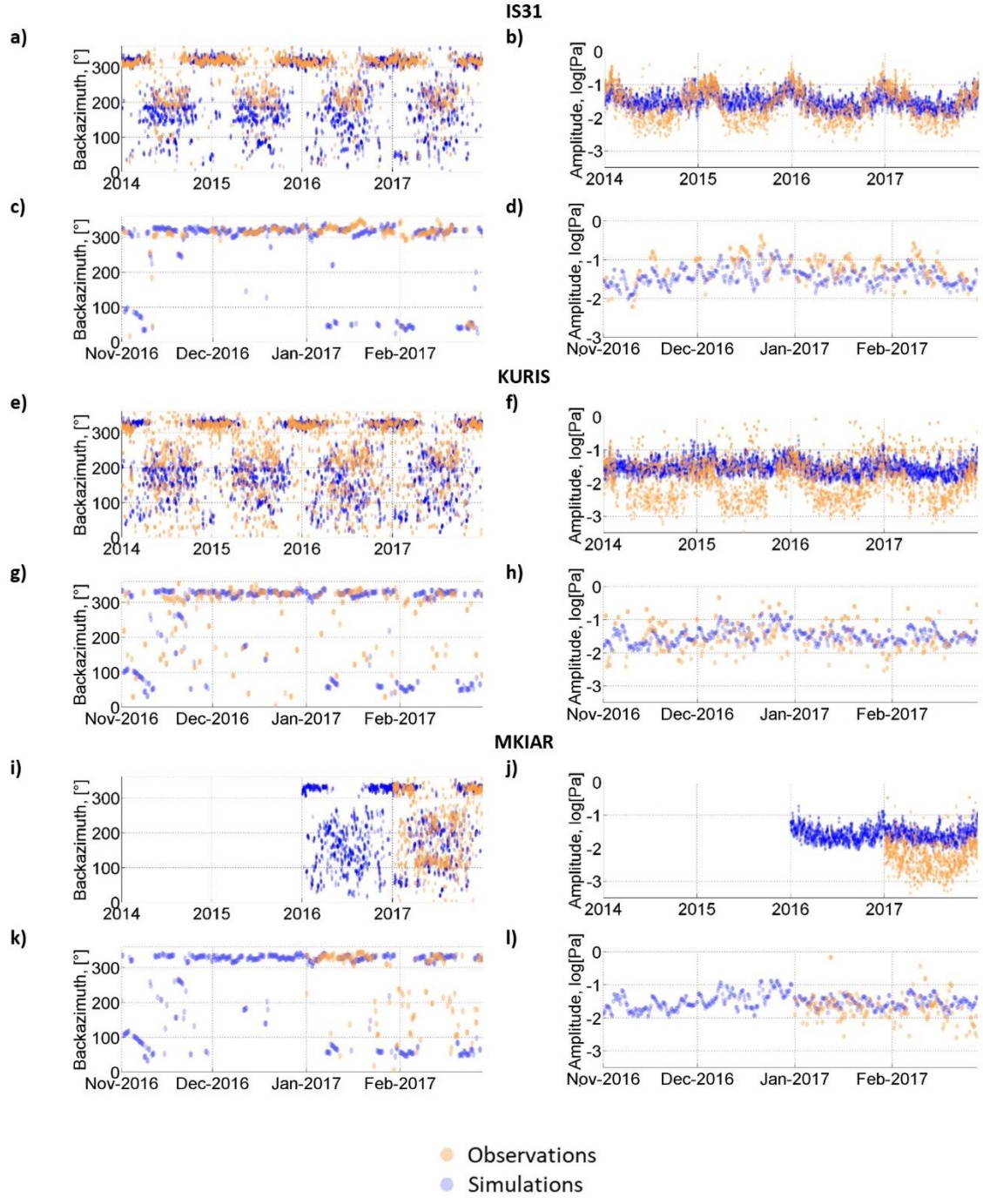

**Figure 2. Time variations of observed back-azimuths and amplitudes of microbaroms at IS31 (a-d), KURIS (e-h), and MKIAR (i-l), with a time resolution of 6 hours from January 1, 2014 to December 31, 2017 (orange circles). Blue circles denote simulated values. Details at IS31 (c,d), KURIS (g,h) and MKIAR (k-l).**

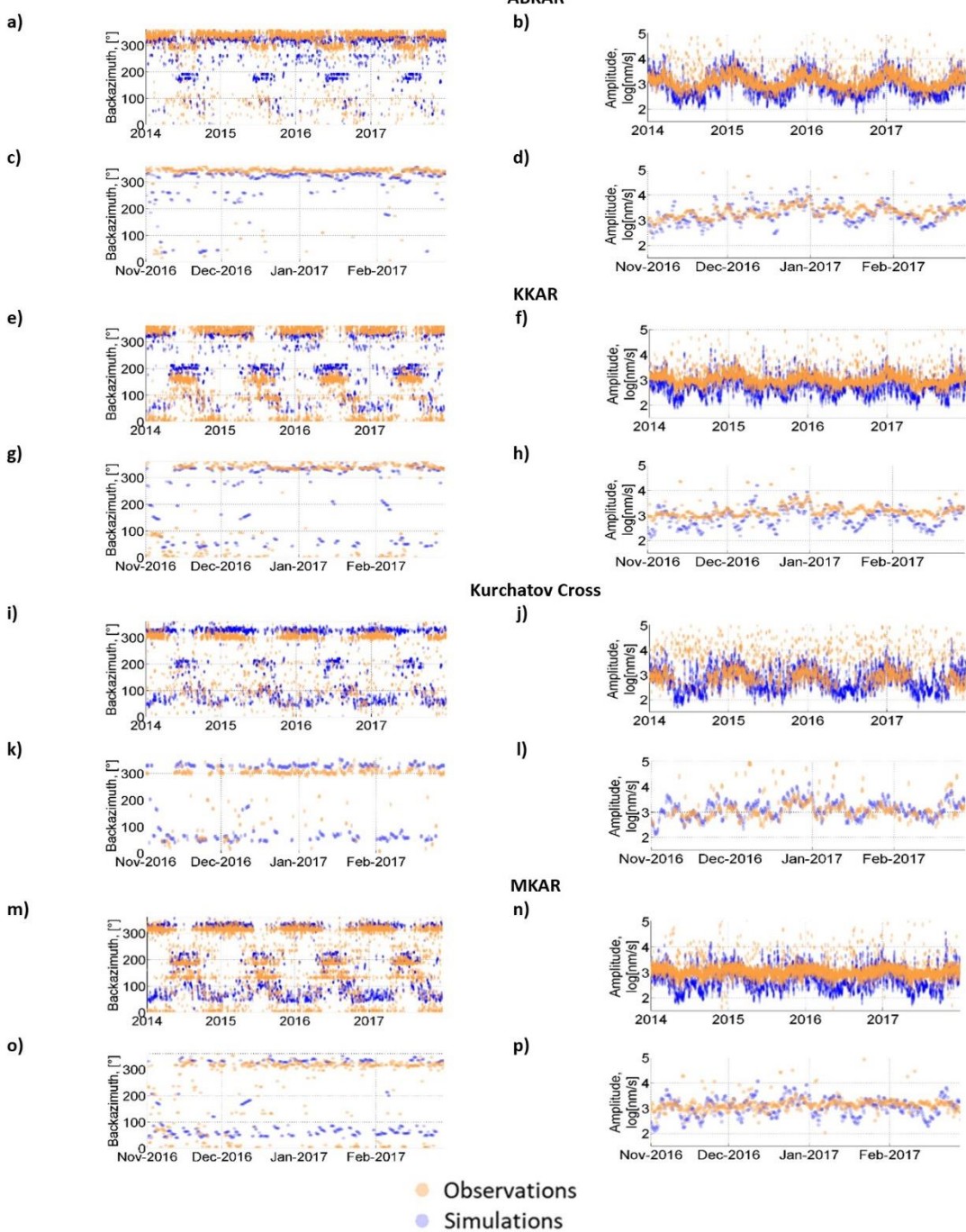

**Figure 3. Same as Figure 2 at ABKAR (a-d), KKAR (e-h), Kurchatov Cross (i-l), and MKAR (m-p).**

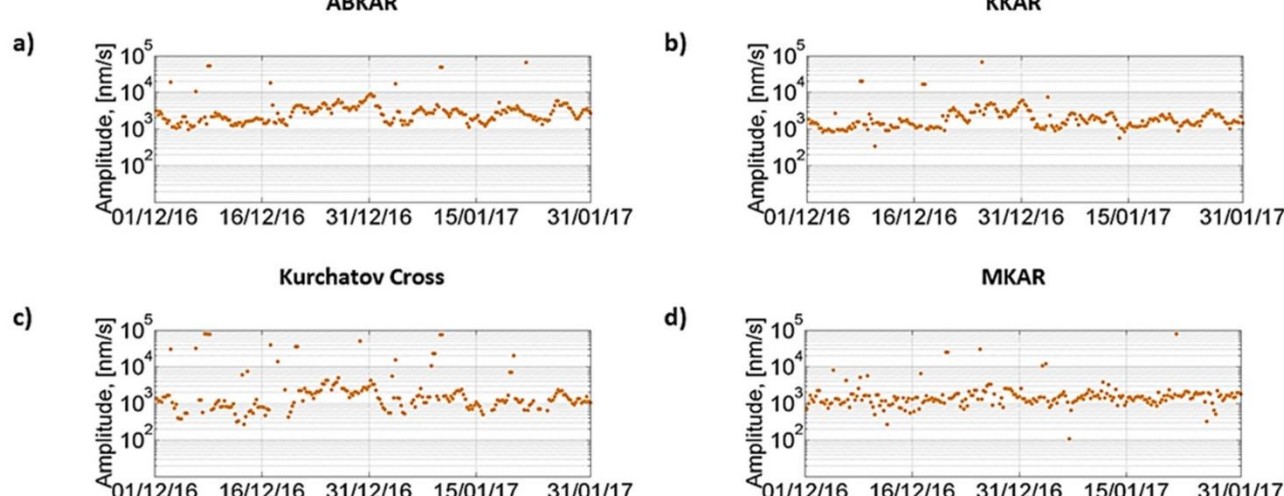

**Figure 4. Dominant amplitude of microseisms in the 0.1-0.4 Hz band detected at ABKAR (a), KKAR (b), Kurchatov Cross (c), and MKAR (d) arrays from December 1, 2016 to January 31, 2017.**

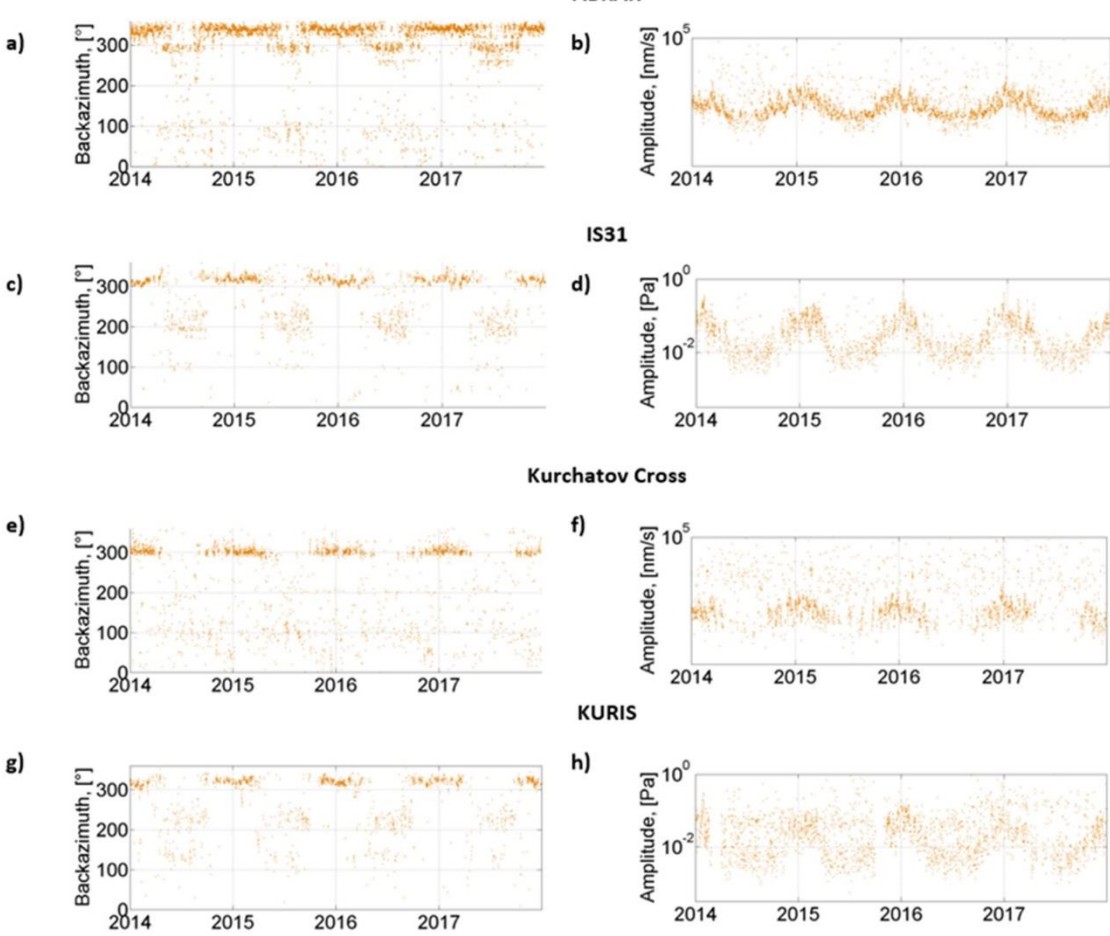

**Figure 5.** Comparison of the observed back-azimuths and amplitudes at ABKAR (a,b) and IS31 (c,d), 230 km apart, and collocated Kurchatov Cross (e,f) and KURIS (g,h) arrays.

**Table 1. Uncertainties of azimuth and apparent velocity estimates.**

| Parameter | IS31 | KURIS | MKIAR | ABKAR | KKAR | MKAR | Kurchatov Cross |
|---|---|---|---|---|---|---|---|
| Horizontal velocity, m/s | 340 | 340 | 340 | 3000 | 3000 | 3000 | 3000 |
| δΘ (°) | 0.55 – 0.74 | 2.05 – 2.34 | 0.58 – 0.67 | 4.89 – 5.64 | 5.14 – 6.30 | 4.55 – 6.84 | 0.48 – 0.49 |
| δV (m/s) | 3.8 – 4.4 | 12 - 14 | 3.5 – 3.9 | 250 – 290 | 270 – 330 | 220 - 380 | 25 – 26 |

**Table 2. Estimations of the prediction quality for microbarom amplitudes and azimuths.**

| Station | Long term observation period | $S_{corr\_Az}$ | $S_{corr\_Amp}$ | Observation period on winter | $S_{corr\_Az}$ | $S_{corr\_Amp}$ | Observation period on summer | $S_{corr\_Az}$ | $S_{corr\_Amp}$ |
|---------|------------------------------|----------------|-----------------|------------------------------|----------------|-----------------|------------------------------|----------------|-----------------|
| IS31 | 2014 – 2017 | 0.61 | 0.39 | Dec 2016 – Feb 17 | 0.76 | 0.53 | Jun 17 – Aug 17 | 0.44 | 0.26 |
| KURIS | 2014 – 2017 | 0.52 | 0.23 | Dec 2016 – Feb 17 | 0.82 | 0.58 | Jun 17 – Aug 17 | 0.16 | 0.18 |
| MKIAR | Sep 16 – Dec 17 | 0.62 | 0.5 | Dec 2016 – Feb 17 | 0.82 | 0.66 | Jun 17 – Aug 17 | 0.34 | 0.39 |

**Appendix A. Instrument responses.**

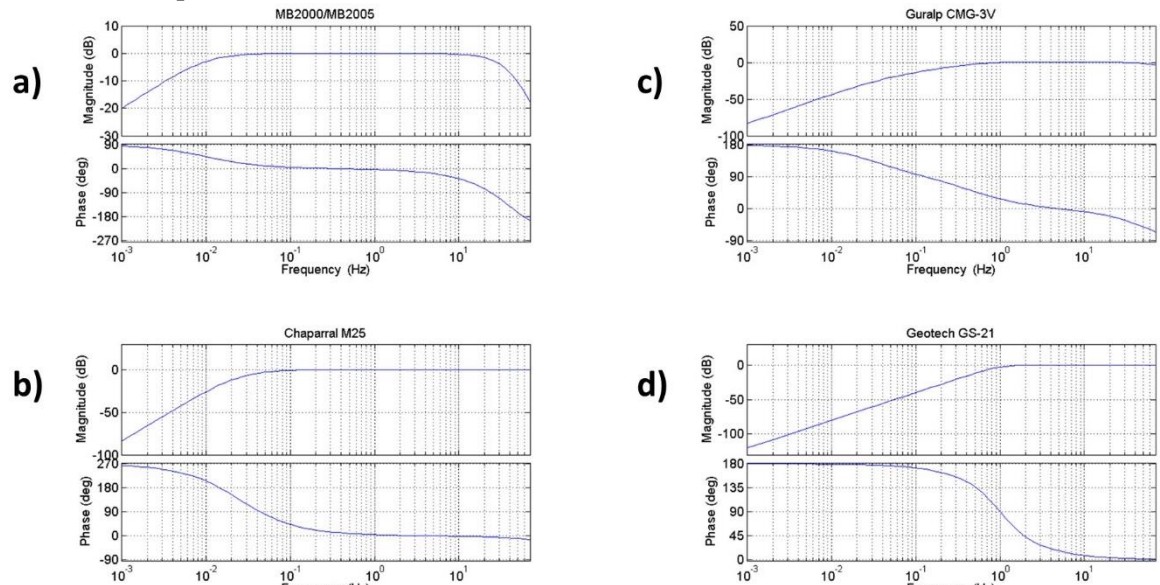

**Figure A1.** Normalized frequency response of the a) MB2000 and MB2005, b) Chaparral M25 microbarometers, c) Guralp CMG-3V, and d) Geotech GS-21 seismometers.

**Table A1. Description of infrasound and seismic arrays**

| Array | Sensor | Response in units lookup | Digitizer | Sampling frequency, Hz |
|---|---|---|---|---|
| IS31 | MB2000 | Pa | DASE Aubrac | 20 |
| KURIS | MB2005 | Pa | Guralp CMG-DM24S6EAM | 20 |
| MKIAR | Chaparral M25 | Pa | Science Horizons AIM24 | 40 |
| ABKAR, KKAR, MKAR | Geotech GS-21 | m/s | Science Horizons AIM24 | 40 |
| Kurchatov Cross | Guralp CMG 3-V | m/s | Nanometrics Europa-T | 40 |

**Appendix B. Noise spectra.**

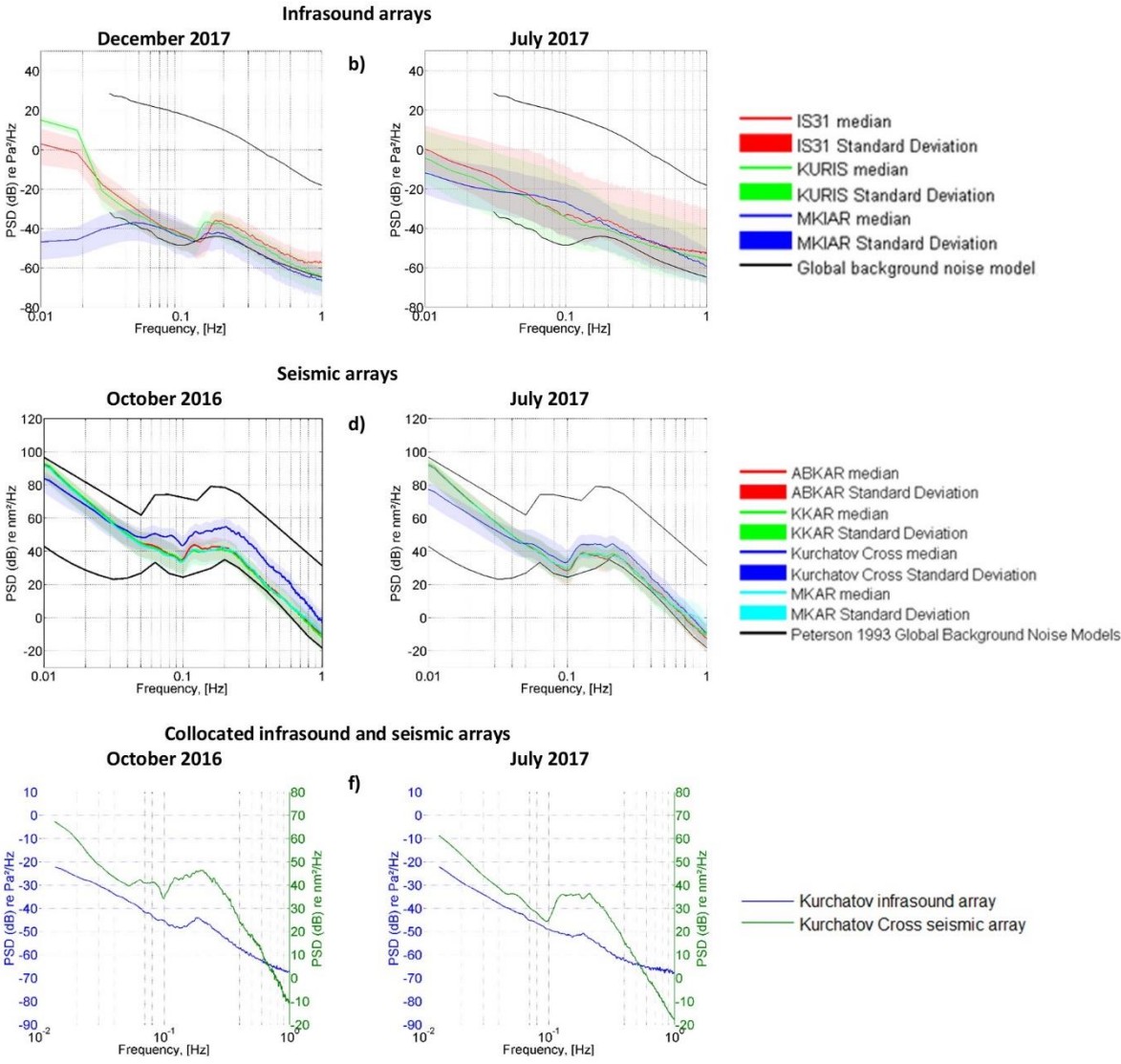

Figure B1. PSD noise spectra at infrasound arrays (a,b) and seismic arrays (c,d). Comparison of noise spectra at collocated KURIS and Kurchatov Cross arrays.

**Appendix C. The distribution of the epicentres of the predicted microbarom sources.**

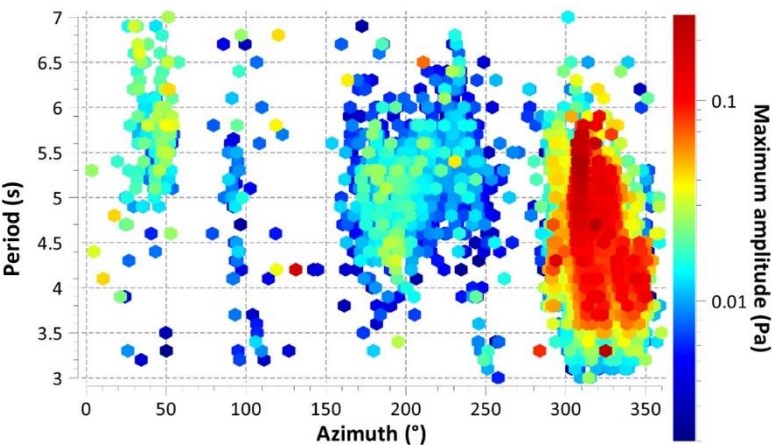

**Figure C1. Signal periods versus back-azimuths at IS31 observations in 2017. The amplitude is color coded (in Pa).**

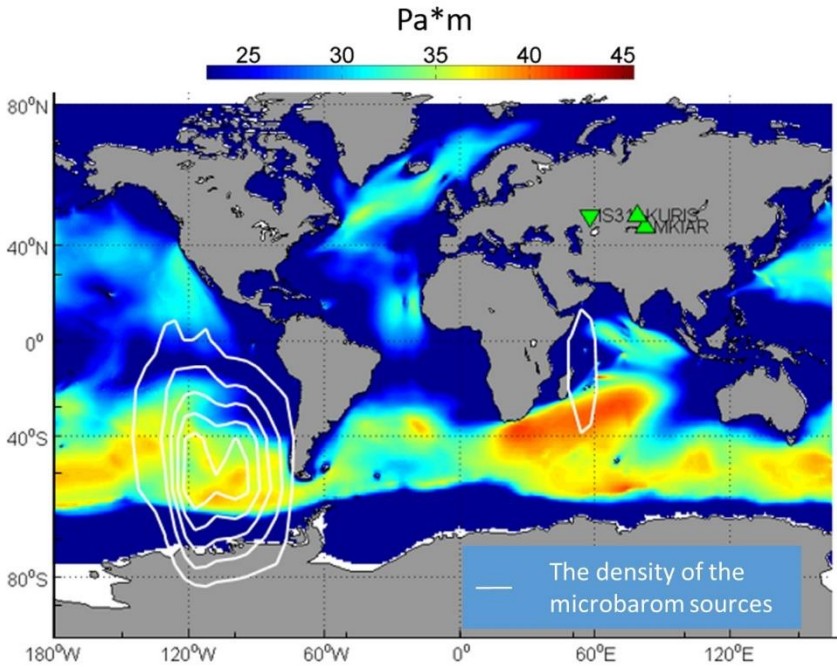

**Figure C2. Spatial distribution of the epicentres of microbarom sources in July-August 2017. White contours represent the density of the microbarom source locations obtained via cross-bearing using detections at IS31, KURIS and MKIAR, during same time periods. At each station, back-azimuths are daily averaged.** 25

**Appendix D. Comparison of backazimuths at collocated seismic and infrasound arrays.**

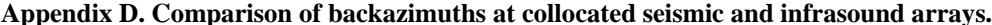

**Infrasound arrays**

a)
**Entire year 2017**

b)
**Winter months 2017**

c)
**Summer months 2017**

— IS31 observed
--- IS31 modeled
— KURIS observed
--- KURIS modeled
— MKIAR observed
--- MKIAR modeled

**Seismic arrays**

d)
**Entire year 2017**

e)
**Winter months 2017**

f)
**Summer months 2017**

— ABKAR observed
--- ABKAR modelled
— Kurchatov Cross observed
--- Kurchatov Cross modelled
— KKAR observed
--- KKAR modelled
— MKAR observed
--- MKAR modelled

**Figure D1. Azimuthal distribution of infrasound detections throughout 2017 (a), from December 1, 2016, to February 28, 2017 (b), and from June 1 to August 31, 2017 (c). Azimuthal distribution of seismic detections throughout 2017 (d), from December 1, 2016, to February 28 (e), 2017, and from June 1 to August 31, 2017 (f).** 26