# Peer review of "Characterizing the oceanic ambient noise as recorded by the dense seismo-acoustic Kazakh network"

_Solid Earth, 2020_

## Short Comment (SC1) · 1 Apr 2020

The presentation "Similarities and differences of microseism and microbarom source regions reconstructed from the seismo-acoustic Kazakhstani network" is available online at the https://meetingorganizer.copernicus.org/EGU2020/EGU2020-2965.html . It covers the same issues as the paper and also some advantages in the microseism simulation.

---

## Referee Comment (RC1) · Anonymous Referee #1 · 20 May 2020

General Comments

In this manuscript authors study the global ocean ambient noise by means of a dense seismo-acoustic network in a multi-year time interval. They present three year of detected microbarom and microseism signals, in order to track and study the temporal variability of ambient noise originating from the two hemispheres. They carried out data analyses and modelling by applying methods and techniques already known in literature and explained and cited them well. Authors modelled microbarom source model and compared the simulated directions and amplitudes with the observed ones. They obtained a good correlation as regarding azimuths, and some discrepancies in

amplitudes, which they discuss into the text. They were able to locate sources of low-frequency signals, and individuated the further studies needed to improve this kind of research. Overall, I think that the manuscript yields a good contribution to the literature in the topic of global ambient noise inasmuch they use coupled seismic and infrasound data, recorded during a three-year long period. I have a few specific comments, but I think that they can be solved by the authors. Results and interpretations are structured in a well way, beside the introduction section, which is, in my opinion, a little bit too long, with respect to the other sections.

Specific Comments

In data processing, when you work on amplitude and phase of the signal, the frequency response of both seismic and infrasonic sensors is a crucial point. This is especially true in this work, where frequency range of signals that you are analysing, is at the bound of the linear frequency response of some of the used sensors. As you show in Figures 3 and 6, you hold the frequency response curve of Microbarometers MB2000, and GS-21 and CMG-3V seismic sensors, I was wondering, did you correct the signal for the frequency response?

I order to make the overall structure of the paper clearer, I believe that few paragraphs of the introduction section could be moved into the method section (for example lines 34-36 and lines 69-76), or in the discussion section (e.g. lines 36-40). I think this would also help to streamline the introduction section.

Also, I think that you could better highlight: i) the contribution of this paper to the literature and ii) the goal of your study and how you try to reach it. You mention that you analysed a long-time interval with simultaneous seismic and infrasonic recording, but it is not so clear if this kind of study, using dense network and lots of data, has been already done in literature.

I suggest that you try to use Rose Diagrams, they could help you representing better the results (both azimuth and frequency of each class) of Figures 13 and 14.

**SED**

Technical corrections

I suggest an English revision, in order to make the manuscript reading more fluent. The bibliography is not cited uniformly throughout the manuscript

Figure 3: x labels are wrong

Figure 7, 8 and 9: Can you put the legend out of the plot? They overwrite the results.

Insert in all the different plot of figures 13 and 14 the letters (e.g. a, b etc).

Line 49-50: Specify which kind of algorithm do you refer to.

Line 53: were

Line 58: measured instead of "measuring"

Line 64-65: "These agreements have been improved using more accurate wind profiles obtained from high resolution LIDAR middle atmospheric sounding". It is not clear that this work has been done by Hupe et al 2018.

Line 66-67: "In this paper, we further extend the approach developed by Hupe at al. (2018) using microbarom recorded by the dense Kazakhstani network". Specify in which sense you extended the work, only in terms of number of station/network?

Line 70: rephrase "For microseisms, the bathymetry strongly affects the source intensity" in "The bathymetry strongly affects the source intensity in microseism modelling"

Line 72: "angles lower than $40°$" instead of "angles lower the $40°$"

Line 74: typing error

Line 83: "as it contains a five seismic and three infrasound arrays". In the abstract and later on into the text and figure 1 you say four seismic arrays.

Line 89: "MKIAR (9 elements), and in Makanchi village" is not clear, MKIAR is in Makanchi village?

Line 92: cut Figure 3

Line 101: cut Figure 5

Line 120-121: Add references

Line 132-134: Understanding this point is difficult in this part of the text, maybe you could move it into the results or discussion section, where you can refer to the figures. Or add here that it would be clarified later into the text.

Line 147-151: Specify that you are describing Fig. 7

Line 155: "amplitude increases from 0.001 to 0.03 Pa" are those average values?

Line 156: I suspect that "repeatable" means replicable.

Line 171: "a decrease in amplitude is observed early January 2017 at all stations." It is difficult to see this decreasing trend.

Line 182-183: "As the used source model was developed for microseisms (Ardhuin et al., 2011), an empirical scaling factor (F = 1:10000) must be applied for comparing the observed to the predicted amplitudes". Could you give further details?

Line 187-190. Could you explain better the reason of the discrepancy? And comment the quantitative estimations of the prediction quality?

Line 192-194: I think it is important here to highlight the further data you analyse in your work, both in terms of time interval and number of stations.

Line 200: Do you mean the comparison between observations and simulations?

Line 200-202: I suggest that you explain better this point.

Line 208: "Simulating microbaroms predicts signals" maybe Simulated microbaroms predict signals.

Line 208-210: Refer to the figures

Line 222: Could you specify here the expanded form of SSW (beside the abstract)?

Line 222-227: As you write here, this topic seems to be one of the findings of the paper.

Line 243-244: This is not reported elsewhere in the text, maybe you could highlight this aspect even in other section, if you believe that it is an important point of your work.
* * *

---

## Author Comment (AC1) · 4 Jun 2020

Thanks a lot for the good overall appraisal of our work. We will do our best to react adequately to the specific comments. It is not a big challenge as we are agreeing with them in most of the cases. The referee is absolutely wright that: "In data processing, when you work on the amplitude and phase of the signal, the frequency response of both seismic and infrasonic sensors is a crucial point. This is especially true in this work, where the frequency range of signals that you are analyzing, is at the bound of the linear frequency response of some of the used sensors. As you show in Figures 3 and 6, you hold the frequency response curve of Microbarometers MB2000, and GS-

21 and CMG-3V seismic sensors, I was wondering, did you correct the signal for the frequency response?" As most of the sensors used in infrasound investigations are broadband the correction for the frequency response is seldom applied when studying the lion's share of the infrasound signals including microbaroms with the aid of the PMCC detector. The procedure was not applied here. However, for the data of the ABKAR, KKAR, and MKAR it should be used for the purposes of the accurate absolute amplitude discrimination. In the opinion of the authors for that particular investigation, this is not a critical failure. Phase responses of the CMG-21 are stable and the absence of the correction doesn't affect the accuracy of the azimuth determination using PMCC. Surely the absolute amplitudes are measured with a huge error. However, the absolute amplitude determination is not an essential result in this paper, only relative amplitudes are shown. But again the response correction is certainly needed when measuring the absolute amplitudes.

If the referee agreed the next version of the preprint will be presented after all the referee reports come.

---

## Referee Comment (RC2) · Jelle Assink (Referee) · 2 Jul 2020

In this paper, the authors present multiple years of microbarom and microseism observations on a dense array network in Kazachstan. The microbarom observations (time series of back azimuth and amplitude as a function of time) are compared with numerical simulations. The authors claim that additional knowledge can be obtained by considering a network of arrays, instead of single array studies, such as done in earlier work.

The authors suggest that this network can be used to develop synergetic approaches to better constrain microbarom source and evaluate propagation effects.

[Figure]

I much like the approach that the authors take in this study and I believe that this paper would be a great addition to the scientific literature. However, before acceptance there are several issues that should be resolved by the authors. Therefore, I would like to recommend a major revision. In particular, I would like the authors to consider the following critiques:

1. I missed in the abstract and introduction some discussion on the novelty of this study: e.g. what the added value is of a characterisation with seismic and acoustic arrays that are part of a dense network. In the conclusion, the authors claim that analyzing multi-year archives of continuous recordings yields additional information about the spatial and temporal variability of the ambient noise originating from two hemispheres. This is an interesting aspect, but in my opinion the manuscript does not provide sufficient evidence for that.

2. A shortcoming is the lack of microseism predictions. Certainly since these simulations can be produced by the same model. Please add these to a revised manuscript.

3. I also missed a more direct comparison of microseism and microbarom observations, e.g. MKIAR/MKAR and KURIS/Kurchatov

3. Some figures are missing where others are superfluous. In particular, I consider that Figures 1, 2, 4 and 5 could be combined in one single figure. I missed figures that (1) show a map of the distribution and characteristics (amplitude/dominant frequency) of microbarom and microseism sources that are considered in this study (also from the southern hemisphere?) and (2) spectral characteristics of the observations, i.e. Probability Density Functions of the Power Spectral Densities for winter and summer months, for all arrays considered.

4. I would like the authors to address spelling and grammatical errors. I have included a few suggestions and have included a rephrasing occasionally.

5. I would like the authors to discuss the shortcomings in the current method (data

processing, range-independence) in a revised version of the manuscript. In particular, the used array processing method is known to produce biased results when the signal consists of multiple, concurrent sources (the case when studying microbarons).

These, and other minor points can be found in the annotated PDF.

Best regards, Jelle Assink

Please also note the supplement to this comment:
https://se.copernicus.org/preprints/se-2020-8/se-2020-8-RC2-supplement.pdf

[Figure]

**Supplement:**

**Characterizing the global ocean ambient noise as recorded by the dense seismo-acoustic Kazakh network**

Alexandr Smirnov1,2, Marine De Carlo3, Alexis Le Pichon3, Nikolai M. Shapiro2,4,5, Sergey Kulichkov6

1Institute of Geophysical Research, Almaty, 050020, Kazakhstan

[revised manuscript text omitted]

---

## Author Comment (AC2) · 6 Jul 2020

Dear Jelle,

Thanks a lot for the benevolent appraisal of the paper in its current state, for the time spent for the review preparation, and for the constructive suggestions. We are agreeing with your remarks. Only clause 2 seems to be a point for discussion, the details are below. If you don't mind we are ready to start the paper correction as it is specified below for each item:

1. I missed in the abstract and introduction some discussion on the novelty of this

[Figure]

study: e.g. what the added value is of a characterization with seismic and acoustic arrays that are part of a dense network.

We will add the discussion.

In the conclusion, the authors claim that analyzing multiyear archives of continuous recordings yields additional information about the spatial and temporal variability of the ambient noise originating from two hemispheres. This is an interesting aspect, but in my opinion the manuscript does not provide sufficient evidence for that.

We will exclude the item from the conclusions.

2. A shortcoming is the lack of microseism predictions. Certainly since these simulations can be produced by the same model. Please add these to a revised manuscript.

You are right, it is possible to produce these simulations using the same source model. The only difference with the microbarom source is the need to take into account the influence of the bathymetry effect. These simulation has been already done, the results were included in our talk at EGU 2020 https://doi.org/10.5194/egusphere-egu2020-2965. This material was not included in the paper in purpose as authors are going to prepare another paper based on the same observational data but dedicated mostly to seismic data. Microseism simulation would be the focal point in this new paper. We propose to clearly mention this effect and evaluate its impact on the modeling

3. I also missed a more direct comparison of microseism and microbarom observations, e.g. MKIAR/MKAR and KURIS/Kurchatov

It is a very good point and we will add the comparison.

4. Some figures are missing where others are superfluous. In particular, I consider that Figures 1, 2, 4 and 5 could be combined in one single figure. I missed figures that (1) show a map of the distribution and characteristics (amplitude/dominant frequency) of microbarom and microseism sources that are considered in this study (also from the southern hemisphere?) and (2) spectral characteristics of the observations, i.e.

[Figure]

Probability Density Functions of the Power Spectral Densities for winter and summer months, for all arrays considered.

We will change the figures and add the map and the PDFs. As the first referee also suggested the changes we will try to find some reasonable compromise.

5. I would like the authors to address spelling and grammatical errors. I have included a few suggestions and have included a rephrasing occasionally.

This is the most difficult point! We will do our best and thanks again for your editing.

6. I would like the authors to discuss the shortcomings in the current method (data processing, range-independence) in a revised version of the manuscript. In particular, the used array processing method is known to produce biased results when the signal consists of multiple, concurrent sources (the case when studying microbarons).

We will add the discussion.

Kind regards

Alex

---

## Author Response (AR1)

| # | Comments from Referees #1 | Author's response | Author's changes in manuscript |
|---|---|---|---|
| 1 | In data processing, when you work on amplitude and phase of the signal, the frequency response of both seismic and infrasonic sensors is a crucial point. This is especially true in this work, where frequency range of signals that you are analysing, is at the bound of the linear frequency response of some of the used sensors. As you show in Figures 3 and 6, you hold the frequency response curve of Microbarometers MB2000, and GS-21 and CMG-3V seismic sensors, I was wondering, did you correct the signal for the frequency response? | As the sensors used in infrasound investigations are broadband the correction for the frequency response is not applied applied when prior PMCC processing. For the data of the ABKAR, KKAR, and MKAR, it is assumed that the frequency response of all sensors of a same array are identical. As PMCC is a correlation based method, identical phase shift should not affect the detection. However, to get reliable amplitude measurements, should be corrected for the instrumental response.  For the purpose of this study, , this is not a critical point as the phase responses of the CMG-21 are stable and the absence of the correction doesn't affect the accuracy of the azimuth determination using PMCC. Surely the absolute amplitudes are measured with a large error. However, the absolute amplitude determination is not essential, only relative amplitudes are  compared with modeling results. | |
| 2 | I order to make the overall structure of the paper clearer, I believe that few paragraphs of the introduction section could be moved into the method section (for example lines 34-36 and lines 69-76), or in the discussion section (e.g. lines 36-40). I think this would | Accepted | Lines 34-36 and lines 69-76 are moved to the methods section. Lines 36-40 are moved to the discussion section. |

| | | | |
|---|---|---|---|
| | also help to streamline the introduction section. | | |
| 3 | you could better highlight: i) the contribution of this paper to the literature and ii) the goal of your study and how you try to reach it. You mention that you analysed a long-time interval with simultaneous seismic and infrasonic recording, but it is not so clear if this kind of study, using dense network and lots of data, has been already done in literature. | To our knowledge, multi-year ambient noise comparisons between co-located seismic and infrasound sensors have not been performed before. | One sentence has been added in the introduction. |
| 4 | I suggest that you try to use Rose Diagrams, they could help you representing better the results (both azimuth and frequency of each class) of Figures 13 and 14. | Accepted | Figures 13 and 14 are reshaped as the rose diagrams (the new numbers are 15 and 16). |
| 5 | I suggest an English revision, in order to make the manuscript reading more fluent. | Accepted | The English has carefully been revised |
| 6 | The bibliography is not cited uniformly throughout the manuscript | Accepted | TODO Citations has carefully been checked throughout the paper |
| 7 | Figure 3: x labels are wrong | Accepted | Figure 3 is reconstructed altogether with the labels. |
| 8 | Figure 7, 8 and 9: Can you put the legend out of the plot? They overwrite the results. | Accepted | The legends are moved out (new numbers are 8, 9 and 10). |
| 9 | Insert in all the different plot of figures 13 and 14 the letters (e.g. a, b etc). | Accepted | The letters are inserted (new numbers are 15 and 16). |
| 10 | Line 49-50: Specify which kind of algorithm do you refer to. | Accepted | This has been clarified |
| 11 | Line 53: were | Accepted | |
| 12 | Line 58: measured instead of "measuring" | Accepted | |
| 13 | Line 64-65: "These agreements have been improved using more accurate wind profiles obtained from high resolution LIDAR middle atmospheric sounding". It is not clear that this work has been done by Hupe et al 2018. | Accepted | The reference to the Hupe et al 2018 is inserted. |
| 14 | Line 66-67: "In this paper, we further extend the approach developed by Hupe at al. (2018) using microbarom recorded by the dense Kazakhstani network". Specify in which sense you extended the work, only in terms of number of station/network? | Accepted | This has been clarified |
| 15 | Line 70: rephrase "For microseisms, the bathymetry strongly affects the source intensity" | Accepted | The bathymetry strongly affects the source intensity in microseism modelling |

| | | | |
|---|---|---|---|
| | in "The bathymetry strongly affects the source intensity in microseism modelling" | | |
| 16 | Line 72: "angles lower than 40" instead of "angles lower the 40" | Accepted | |
| 17 | Line 74: typing error | Corrected | |
| 18 | Line 83: "as it contains a five seismic and three infrasound arrays". In the abstract and later on into the text and figure 1 you say four seismic arrays. | The BVAR was not shown at Figure1. There are 5 arrays now. | |
| 19 | Line 89: "MKIAR (9 elements), and in Makanchi village" is not clear, MKIAR is in Makanchi village? | Corrected | The infrasound network consists of the IMS infrasound station IS31 located in north-west Kazakhstan (2.1 km aperture, 8 elements), two national arrays of 1 km aperture: KURIS (4 elements) near the Kurchatov, MKIAR (9 elements) near the Makanchi village |
| 20 | Line 83: "as it contains a five seismic and three infrasound arrays". In the abstract and later on into the text and figure 1 you say four seismic arrays. | Corrected | Five seismic |
| 21 | Line 92: cut Figure 3 | Corrected | It is Figure 1 now. |
| 22 | Line 101: cut Figure 5 | Corrected | It is Figure 1 now. |
| 23 | Line 120-121: Add references | | |
| 24 | Line 132-134: Understanding this point is difficult in this part of the text, maybe you could move it into the results or discussion section, where you can refer to the figures. Or add here that it would be clarified later into the text. | Accepted and deleted as the same is stated again in the discussion section. | |
| 25 | Line 147-151: Specify that you are describing Fig. 7 | Accepted | The reference to Picture 8 is added |
| 26 | Line 155: "amplitude increases from 0.001 to 0.03 Pa" are those average values? | No, these are the maximal values. | the maximal signal amplitude increases from 0.001 to 0.03 Pa. |
| 27 | Line 156: I suspect that "repeatable" means replicable. | No, actually it is the repetitive. | "with repetitive seasonal variations" |
| 28 | Line 171: "a decrease in amplitude is observed early January 2017 at all stations." It is difficult to see this decreasing trend. | The amplitude scales are uniform now. The effect must be more visible. | |
| 29 | Line 182-183: "As the used source model was developed for microseisms (Ardhuin et al., 2011), an empirical scaling factor (F = 1:10000) must be applied for comparing the observed to the predicted amplitudes". Could you give further details? | Accepted | This has been clarified |

| 30 | Line 187-190. Could you explain better the reason of the discrepancy? And comment the quantitative estimations of the prediction quality? | Accepted | This has been clarified |
|----|----|----|----|
| 31 | Line 192-194: I think it is important here to highlight the further data you analyse in your work, both in terms of time interval and number of stations. | Accepted | This has been mentioned in the discussion |
| 32 | Line 200: Do you mean the comparison between observations and simulations? | No, we mean but not the comparison but both of them do. | Observations as well as simulations, show large temporal variations in the dominating microbarom source regions explained by the seasonal reversals of the prevailing stratospheric winds, which in turn, cause the migration of storm activity area to the winter hemisphere (Stutzmann et al., 2012). |
| 33 | Line 200-202: I suggest that you explain better this point. | Accepted | This has been clarified |
| 34 | Line 208: "Simulating microbaroms predicts signals" maybe Simulated microbaroms predict signals. | Accepted | |
| 35 | Line 208-210: Refer to the figures | The reference is added | "observed only at IS31 and MKAR, Figure 15 c." |
| 36 | Line 222: Could you specify here the expanded form of SSW (beside the abstract)? | Accepted | |
| 37 | Line 222-227: As you write here, this topic seems to be one of the findings of the paper. | Yes, as stated in the text | |
| 38 | Line 243-244: This is not reported elsewhere in the text, maybe you could highlight this aspect even in other section, if you believe that it is an important point of your work. | Accepted | This result is expanded in the discussion. |

| # | Comments from Referees #2 | Author's response | Author's changes in manuscript |
|---|---|---|---|
| 1 | I missed in the abstract and introduction some discussion on the novelty of this study: e.g. what the added value is of a characterisation with seismic and acoustic arrays that are part of a dense network. | Accepted | To our knowledge, systematic comparisons between observed and predicted microseisms and microbaroms were not carried out. This paper confirms the pioneer findings of Donn by considering several years of observations at a dense regional seismo-acoustic network. Such comparisons confirm a common source mechanism of seismic and acoustic ocean ambient noise while highlighting the influence of long-range atmospheric propagation on microbarom prediction. This has been clarified in the text. |
| 2 | In the conclusion, the authors claim that analyzing multiyear archives of continuous recordings yields additional information about the spatial and temporal variability of the ambient noise originating from two hemispheres. This is an interesting aspect, but in my opinion the manuscript does not provide sufficient evidence for that. | Accepted | The issue is excluded from the conclusion |
| 3 | A shortcoming is the lack of microseism predictions. Certainly since these simulations can be produced by the same model. Please add these to a revised manuscript. | Accepted | The microseism predictions are inserted. This is done via the addition of Figures 12 – 14. The predictions calculation is described in the method section. The results are discussed in the section results and discussions. |
| 4 | I also missed a more direct comparison of microseism and microbarom observations, e.g. MKIAR/MKAR and KURIS/Kurchatov | Accepted | The comparison is done via the placement of figures 19 and 20. The comparison is described in the discussion section. |
| 5 | Figures 1, 2, 4 and 5 could be combined in one single figure. | Accepted | The figures are combined into the new version of figure 1. |
| 6 | missed figures that: (1) show a map of the distribution and characteristics (amplitude/dominant frequency) of microbarom and microseism sources that are considered in this study (also from the southern hemisphere?) | Accepted | These maps are inserted as figures 6 and 7 for microbarom and microseism sources accordingly. The description is done in the methods section. |
| 7 | missed figures that: (2) spectral characteristics of the observations, i.e. Probability Density Functions of the Power Spectral Densities for winter and summer months, for all arrays considered. | Accepted | The PSDs are shown in picture 4. The description is at the observation network section. |

| | | | |
|---|---|---|---|
| 8 | I would like the authors to address spelling and grammatical errors. I have included a few suggestions and have included a rephrasing occasionally. | Accepted and checked where possible. | |
| 9 | I would like the authors to discuss the shortcomings in the current method (data processing, range-independence) in a revised version of the manuscript. In particular, the used array processing method is known to produce biased results when the signal consists of multiple, concurrent sources (the case when studying microbarons). | Accepted | This limitation is now clearly addressed in the conclusion. |
| 10 | Line 10 In the asbtract:
- describe the results in more detail
- what are the differences between modeling and observations?
- not much emphasis on the microseism signals
- what is the broader perspective of this characterization using seismic and acoustic arrays? | Accepted | The outcomes of this study were reformulated (see point 1). |
| | Line 14 that are part | Accepted | |
| 11 | Line 25 Microseism and microbarom modeling techniques were preceded by years of observations.
Add a paragraph on the history observations of microseisms and microbaroms to introduce the topic of this paper; e.g. the work by Benioff and Gutenberg.
I also missed a review of the work by Donn and Rind from the 1970s which is important to mention. Rind also published a study in 1980 discussing the joint observation of microseisms and microbaroms. | Accepted | Reviews of pioneer work on microseism microbaroms have been added in the introduction. |
| 12 | Line 29 Microbaroms and microseisms are not only generated in the middle of the ocean by opposing wavetrains but also near coastlines (coastal reflection). Discuss this | Accepted | See point 11. |
| 13 | Line 29 Introduce primary and secondary microseisms. The primary microseisms do not have a counterpart in the atmosphere. Discuss why. | Accepted | See point 11. |

| 14 | Line 29 This study would certainly benefit from a Figure that shows probability density functions of the power spectral density of co-located microbarom and microseism arrays, in particular KURIS/Kurchatov and MKAR/MKIAR | Accepted | The comparison is shown by new figures (19 and 20) and described in the discussion section. |
|----|----|----|----|
| 15 | Line 31 Explain how the Rayleigh waves are predicted by the acoustic pressure source. Are there also homogenous P-waves generated? | Accepted | One reference is added. |
| 16 | Line 32 I would move this paragraph after (KNMI network, Evers and Haak 2001). | Accepted | The paragraph moved to the method section. |
| 17 | Line 35 this is a major assumption. Why is the impedance condition not taken into account? | Accepted | We agree. The bathymetry effect plays an important role when calculating the microseism source intensity as resonance effects occur leading to a modulation of the pressure fluctuations at the ocean bottom. This effect has been modeled using compressible amplification factor of Stutzmann et al. (2012). This has been clarified in the manuscript. |
| 18 | Line 41 I would move this paragraph after (KNMI network, Evers and Haak 2001). | Accepted | The paragraph is moved to the recommended place. |
| 19 | Line 46 needs an introduction how microbaroms relate to microseisms | Accepted | See point 11. |
| 20 | Line 49 Include a paragraph to review recent articles on high-resolution beamforming methods for microseisms (Gal et al., 2016) and microbaroms (Ouden et al., 2020). Such methods are absolutely needed to resolve the complex infrasonic wavefield at microseism/microbarom frequencies as classical Bartlett and PMCC type methods fall short due to biases. | Accepted | This has been addressed in the conclusion. See point 9. |
| 21 | Line 53 Microseism modeling | Accepted | Microseism modelling was |
| 22 | Line 57 have been | Accepted | Have been |
| 23 | Line 57 Going back to the Garcés et al., 2004 study, I don't see any modeled microbaroms. The authors did compare observations with wind directions at different altitudes. | Accepted | The sentence has been deleted. |
| 24 | Line 61 operational | Accepted | |
| 25 | Line 64 Northern | Accepted | |
| 26 | Line 64 It is not at all clear that high-resolution atmospheric sounding methods | Accepted | This has been suppressed. |

| | | | |
|---|---|---|---|
| | would improve microbarom observations; as the wavelengths of microbaroms are large (1.7 km), it is only sensitive to the larger scale structure that is well captured in atmospheric models. Moreover, the Hupe study does not provide evidence that the agreement actually gets better. I would like the authors to address this. | | |
| 27 | Line 66 explain how you extend the method and what the point is of this paper. | Accepted | This has been clarified. |
| 28 | Line 68 includes both seismic and infrasound arrays. | Accepted | both seismic and infrasound arrays |
| 29 | Line 70 In contrast to the microseism work, the influence of bathymetry on microbaroms has only been studied theoretically. There is no data yet to support this claim. Please describe this as such. Something like, "A recently modeling study by De Carlo (2020) suggests that bathymetry has negligible impact on microbarom source strength in contrast to predictions from the model by Waxler (2007)". | Accepted | |
| 30 | Line 73 propagation angles through the atmosphere | Accepted | |
| 31 | Line 74 already stated above | Accepted | The sentence has been deleted. |
| 32 | Line 75 Rewrite. This sentence reads as if microbaroms are not affected by geometrical spreading and attenuation but by "the strong spati-temporal variation of the middle atmosphere". Indeed, in both cases the propagation conditions (propagation path with its geometrical spreading and attenuation) are determined by the medium properties (which are temperature/wind for infrasound and elastic parameters for seismic waves). In the case of seismic waves, these medium properties do not vary in time. | Accepted | This has been clarified. |
| 33 | Line 75 Microseism modeling should also be considered in this paper; it would make the story much stronger as both waveform technologies are compared. | Accepted | The microseism predictions is considered. This is done via the addition of Figures 12 – 14. The predictions calculation is described in the methods section. The results are discussed in the section results and discussions. |

| 34 | Line 80 After reading this section, it is not clear to me what this paper adds to the existing knowledge. This should be stated, for example in the forelast paragraph. | | This has been clarified. See point 1. |
|---|---|---|---|
| 35 | Line 80 studies | Accepted | |
| 36 | Line 80 I count four in Figure 1? From reading the manuscript, it looks like you have not plotted BVAR. | The BVAR is added to figure 1. There are 5 now. | Figure 1 is changed |
| 37 | Line 85 not relevant for scientific paper. | Accepted | The sentences are deleted. |
| 38 | Line 85 and | Accepted | and |
| 39 | Line 91 which type? | Accepted | Model 25 |
| 40 | Line 92 double | Accepted | Deleted |
| 41 | Line 93 This station is not used so it is irrelevant to mention. | Accepted | Deleted |
| 42 | Line 94 discriminating between | Accepted | between |
| 43 | Line 97 consists of | Accepted | Consists of |
| 44 | Line 98 in a | Accepted | In a |
| 45 | Line 98 It appears to me that Figures 1, 2, 4, and 5 can be combined to show the overview map and the array layouts in one single figure. This will reduce the large number of figures in this paper (15). Alternatively, the seismic arrays can be combined similarly as Figure 2 for infrasound. It would be useful to plot the seismic arrays in Cartesian coordinates so that the array layouts can easily be compared to the infrasound arrays (Figure 2). | Accepted | The figures are combined into the new version of figure 1. |
| 46 | Line 98 Manufacturer? | Accepted | Guralp CMG-3v, also Geotech Instruments GS-21 at the line 101 |
| 47 | Line 99 Figure 6 suggests that it is a broadband sensor.. why would it be at the edge of the response? | Accepted | This has been clarified. |
| 48 | Line 99 sensor's | Accepted | |
| 49 | Line 100 The ABKAR | Accepted | |
| 50 | Line 101 Figure 5 | Accepted | Deleted |
| 51 | Line 101 a flat | Accepted | |
| 52 | Line 103 Perhaps better to show the frequency response with a logarithmic x-axis and include higher frequencies so that you can show the behavior of the GS-21 sensors above 1 Hz. | Accepetd | Figures 2 and 3 have been modified |

| | | | |
|---|---|---|---|
| | More realistic / smoother response plots will result from more points in the calculation of the response curves. | | |
| 53 | Line 107 This point is an important added value of the paper and should be brought out in a revised version. Comparisons of co-located seismic and acoustic arrays would be novel. | Accepted | This has been highlighted in the introduction. |
| 54 | Line 113 Similar processing configurations should be used, including the same frequency band and window lengths. | Accepted | This setting for seismic processing has been chosen as it yields more stable detection results compared with the log-scaling configuration used for microbaroms. Further studies are needed to investigate the most suitable processing scheme as addressed in the conclusion. |
| 55 | Line 116 The authors should include array response functions and estimate for all arrays so that the resolution can be estimation from the lobe width. | Accepted | This has been addressed in the conclusion as further work to estimate realistic uncertainties in the wave parameter estimates. |
| 56 | Line 118 Can you explain the calculations more? The errors are quite dependent on SNR conditions and it may be that the current estimates are slightly optimistic, if for example a sigma  tau of 0.05 s is chosen (following Szuberla & Olson, 2004). Another way of looking at this is to consider the array response function and consider the lobe width. | Accepted | This has been mentioned. See point above. |
| 57 | Line 123 Microbarom sources are | Accepted | |
| 58 | Line 124 which frequency band is considered? | | This has been clarified. |
| 59 | Line 127 scenarios | Accepted | |
| 60 | Line 127 using the high-resolution forecast (HRES) that is part of ECMWF's Integrated Forecast System (IFS) cycle 38r2 | Accepted | using the high-resolution forecast (HRES) that is part of ECMWF's Integrated Forecast System (IFS) cycle 38r2 |
| 61 | Line 130 Good is qualitative. Can you further quantify this? Within how many standard deviations can the model explain the data, for example? | Accepted | Quantitative measures are given. |
| 62 | Line 131 Results should not be part of the methods section. | Accepted | Deleted |
| 63 | Line 132 This interpretation should be saved for the discussion and not be part of the methods section. | Accepted and deleted as the same is stated again in the discussion section. | |

| | | | |
|---|---|---|---|
| | Moreover, it should be supported by data, for example by looking at variations in the effective sound speed along the various great circle paths that are studied. It should also be investigated what the differences in distance for a "southern" vs. "N. Atlantic source" would be. | | |
| 64 | Line 134 Looking ahead at the observations, it seems like sources are more distributed in the south. There are two things to consider: (1) From the array processing perspective: As PMCC cannot detect more than one microbarom source per time-window, it could be that the resolution of such microbaroms is limited. This motivates the use of high-resolution methods such as discussed by den Ouden et al., 2020. (2) From the propagation perspective, there could be multiple paths/ducts from which microbarom energy can reach the array, leading to the observations of multiple infrasound sources (e.g., Assink et al., 2014). Thus, the paradigm of only observing propagation down-wind is challenged at microbarom frequencies. | Accepted | As suggested, explanations are given in the conclusion. See also point 20. |
| 65 | Line 143 Separate microbarom and microseism observations in two subparagraphs, this will make it easier to read. | Accepted | The paragraph was separated on two subparagraphs: 1.3.1 Source modeling for microbaroms and 1.3.2 Source modeling for microseisms |
| 66 | Line 143 Start with 2.1.1: microbaroms | Accepted | |
| 67 | Line 144 suggest rephrase: Figures 7 through 9 | Accepted | |
| 68 | Line 145 suggest rephrase: of the dominant microbarom signals for infrasound arrays IS31, KURIS and MKIAR, respectively. | Accepted | of the dominant microbarom signals for infrasound arrays IS31, KURIS and MKIAR, respectively. |
| 69 | Line 145 suggest rephrase: The amplitudes and back azimuths from the dominant microbarom signals are selected from the PMCC bulletins and are plotted as orange dots. | Accepted | The amplitudes and back azimuths from the dominant microbarom signals are selected from the PMCC bulletins and are plotted as orange dots. |

| 70 | Line 147 Save this for the modeling paragraph. | Accepted | Deleted |
|---|---|---|---|
| 71 | Line 148 back azimuths | Accepted | |
| 72 | Line 153 azimuthal ranges of | Accepted | |
| 73 | Line 154 the KURIS | Accepted | |
| 74 | Line 154 shows | Accepted | |
| 75 | Line 155 Is there a reason why amplitudes would not increase to 0.1 Pa? | | We have no explanation. |
| 76 | Line 157 Discuss how these distributions are computed | Accepted | This is the standard deviation around the dominant detected azimuths |
| 77 | Line 158 Suggest to have this in 2.1.2: microseisms | Accepted | |
| 78 | Line 161 rephrase. A 'detection system' is not appropriate. | Accepted | reworded |
| 79 | Line 161 Same. | Accepted | Same |
| 80 | Line 164 Is this related to the larger aperture of Kurchatov Cross and the loss of coherency? Can you identify a shift in frequency from winter to summer? | Accepted | This could be explained by higher noise level or a loss of signal coherency. We don't see clear shift in frequency from summer to winter. |
| 81 | Line 168 could this be related with the southern location of these arrays? | Accepted | This is suggested. |
| 82 | Line 173 can you explain why MKAR shows so much scatter, relatively? | | We have no explanation. |
| 83 | Line 173 Can you explain why the Kurchatov array appears noisier than the other sites? Perhaps because the seismometers are not installed in boreholes? Or is it related to the instrument? The amplitudes also seem higher than the other sites. Can you adjust the vertical scale so that all are equal? | Accepted | The vertical scale is adjusted. |
| 84 | Line 175 The microbaroms simulations have been computed for the sea states around the infrasound arrays, not for the microbarom recordings. Please provide more detail about the computation, which distances are considered? Are very low amplitudes cut off from the computation? There is a large difference between the simulations and observations in the summer at all arrays. | Accepted | The back-azimuths and amplitudes have been calculated for the expected microbarom sources at IS31, KURIS, and MKIAR. The expected distances to the source regions vary with season. For example at IS31, simulations predict in winter three source regions (Figure 6 a); distances to North Atlantic regions range between 3500 to 7000 km while the distance to the North Pacific region is around 7000 km.  In summer, additional microbarom sources are located in the southern hemisphere at distances larger than 11000 km (Figure 6 b). |

| 85 | Line 175 Figures 7 through 9 | Accepted | Figures 8 through 10 |
|---|---|---|---|
| 86 | Line 176 This, in a way, is not so spectacular, given the large distance to this source, making it appear to come from one dominant azimuth. I would discuss that the scatter of microbarom sources tells something about the relative distance to the source. This can also be seen in Den Ouden et al.,2020 (compare for example IS42 with IS48). | | We think that the relation between the scattering and the relative distance to the source is not clear, as large scattering is also noted for the farthermost source regions in the southern hemisphere. |
| 87 | Line 178 Can you quantify the deviation? | Accepted | |
| 88 | Line 182 I note that the model that is used for microbarom modeling could indeed be used to simulate microseisms. Please explain why amplitudes need an emperical factor and do not follow from the physics. Furthermore, does this mean that amplitude is a free parameter? | Accepted | As the used source model was developed for microseisms (Ardhuin et al., 2011), an empirical scaling factor (F = 1:10000) has been applied to account for wave coupling effect in the atmosphere, thus allowing qualitative comparisons between the observed and predicted temporal variations of the microbarom amplitudes. |
| 89 | Line 185 could one not equally argue that summer amplitudes are correct while winter amplitudes are underestimated? | | We believe that microbarom modeling is not correct for long propagation range as atmospheric model taken at the station likely predicts more favorable propagation conditions compared with situation involving waves crossing the equator line. |
| 90 | Line 188 where would these sources be? please further explain. It would be good if a Figure would be devoted to microbarom / microseism sources during summer and during winter. | Accepted | Figures 6 and 7 are devoted to the microbarom and microseism sources accordingly. The descriptions are in the methods section. |
| 91 | Line 191 I would like the authors to discuss the shortcomings in the current method (data processing, range-independence) in a revised version of the manuscript. | Accepted | This study provides a first characterization of the seasonal patterns of microbarom and microseisms recorded by the IGR seismo-acoustic network. The chosen detection algorithm and propagation model offer a good trade-off between low calculation effort and propagation accuracy. Identified shortcoming is the limitation of PMCC to detect overlapping microbarom sources originating from different directions. Furthermore, the approach assuming range-independent atmosphere may lead to erroneous interpretations for situations involving long propagation ranges where significant along-path variability of wind and temperature profiles may occur, in particular when modeling the relative strength of microbarom sources located in different hemispheres. This has been clarified in the manuscript. |

| 92 | Line 194 Explain how the number of detections can be quantified and be directly compared to a simulated value. Are the number of detections averaged over 6 hours to be directly compared with the simulations? What is the role of wind noise? | Rejected, misunderstanding is a result of mistranslation. The translation was improved. | Figure 15 shows the azimuthal distribution of infrasound detections having the maximum amplitudes. |
|---|---|---|---|
| 93 | Line 198 Repeat from above: Looking ahead at the observations, it seems like sources are more distributed in the south. There are two things to consider: (1) From the array processing perspective: As PMCC cannot detect more than one microbarom source per time-window, it is likely that the ability to resolve microbaroms is limited and biased. This motivates the use of high-resolution methods such as discussed by den Ouden et al., 2020. (2) From the propagation perspective, there could be multiple paths/ducts from which microbarom energy can reach the array, leading to the observations of multiple infrasound sources (e.g., Assink et al., 2014). Thus, the paradigm of only observing propagation down-wind is challenged at microbarom frequencies. | Accepted. | We agree with this limitation. This has been clarified in the conclusion. See also points 20 and 64. |
| 94 | Line 206 I suppose this is identified using trace velocity. Could the authors clarify? | Yes, it is true. | These peaks could likely be explained by body and surface seismic phases judging by its trace velocity. |
| 95 | Line 214 Please include information the typical distances to microbarom/microseism sources and a map showing the typical locations of sources that are likely to be detected. | Accepted | The maps are in figure 6 and 7 |
| 96 | Line 228 given the large number of assumptions in this paper, I would be careful with placing the inaccuracy with the ECMWF model. | Accepted | We suppressed this statement. |
| 97 | Line 231 This would rather be a transient signal; how would that lead to such a large mismatch? | Accepted | We suppressed this statement. |
| 98 | Line 237 please provide evidence for this statement. | Accepted | The sentence has been reworded. |

| 99 | Line 239 During minor SSWs, bi-directional conditions may occur which may have strong impacts on the retrieved microbarom signals (see Assink et al., 2014; JGR). | Accepted | This has been added. |
|---|---|---|---|
| 100 | Line 243 but the wavelength is also 10 times larger, so I cannot imagine how the azimuthal errors would decrease. this also assumes that microbaroms and microseisms originate from the same location. | Accepted | The sentence has been suppressed. |
| 101 | Line 389 BVAR is not plotted | Accepted | |
| 102 | Line 411 - Make the vertical size larger so you can see more detail. - Include more tick marks for the back azimuth values. | Accepted | The vertical size is enlarged. A bigger amount of the tick marks are included. |
| 103 | Line 418 - Make the vertical size larger so you can see more detail. - Include more tick marks for the back azimuth values. | Accepted | The vertical size is enlarged. A bigger amount of the tick marks are included. |
| 104 | Line 421 - Make the vertical size larger so you can see more detail. - Include more tick marks for the back azimuth values. | Accepted | The vertical size is enlarged. A bigger amount of the tick marks are included. |
| 105 | Line 427 Can you make the y-axis scale the same? | Accepted | Y-axis scales are made uniform. |
| 106 | Line 432 Can you make the y-axis scale the same? | Accepted | Y-axis scales are made uniform. |

[revised manuscript text omitted]

¶
¶

---

## Referee Report (RR1)

540

[referee-annotated manuscript omitted]

---

## Author Response (AR2)

| # | Comments from Referees #2 | Author's response | Author's changes in manuscript |
|---|---|---|---|
| 1 | I thank the authors for a careful revision of their manuscript and particularly could note improvements to the presentation of the work, both considering the figures and the English of the paper. I reviewed the manuscript again, but still find that this work could benefit from some corrections before it is accepted for publication. | We are very appreciative of the reviewer's constructive comments which help us improving our paper. We believe that we applied all reasonable efforts possible to answer all questions of the reviewer | |
| 2 | In particular, some sections of Section 1 need some major rewriting as some parts are incorrect, incomplete or too convoluted. Where possible, I have indicated this in the annotated manuscript, but I encourage the authors to critically assess the flow of the paper themselves. As there are many figures in this paper, the authors should consider if some can be included as a supplemental or if they can be combined into one. | The introduction section has been revised carefully following reviewer's constructive comments.

The number of pictures has been reduced to five: some pictures were combined, others were included as a supplement. | |
| 3 | In addition, there are several remaining issues with the presentation and interpretation of the data. The presented variation in amplitude seems problematic for both infrasound and seismic data. This is a major issue because amplitude is an important factor and differences between stations are a major result from this paper. For infrasound, the lower end of the amplitude range is below noise curve, which is not possible. For seismic arrays, there are large differences between the arrays that I suspect to be due to not taking the sensor response into account. This should be trivial to fix but important. | All amplitude issues have been resolved. For both infrasound and seismic data, all plots have been fully refurbished by substituting RMS amplitudes to peak-to-peak values. Amplitudes of the seismic data have also been corrected taking into account sensor response which helped improving the presented materials. | |
| 4 | In addition, there remain several major differences predictions and observations that need to be explained. For the seismic arrays, I wonder if some of the differences can be due to slowness-azimuth station corrections not taken into account. | The authors support the idea that source-specific station correction (SSSC) could improve prediction accuracy. However, such effort is out of the scope of this paper. It would take significant additional time to collect enough GT information to construct the SSSCs, which cannot be done in the present study. | |

| | | | |
|---|---|---|---|
| 5 | Finally, what I missed in the manuscript was an effort to localise the microbarom and microseism sources based on the array observations alone. Throughout the paper, several references are made to microbarom sources on the southern hemisphere without providing any evidence from this based on the array data. | Attempt to localize microbarom sources in the southern hemisphere based on the array observations alone has been successful. This is to our knowledge the first evidence of such remote sources. | |
| 6 | Line 1 oceanic | Accepted | Oceanic |
| 7 | Line 1 omit global, it is no global study. | Accepted | Deleted |
| 8 | Line 11 Reword to bring out the main message better:  In this study, the dense seismo-acoustic network of the Institute of Geophysical Research (IGR), National Nuclear Center of the Republic of Kazakhstan is used to characterize global ocean ambient noise. As the monitoring facilities are co-located, this allows for a joint seismo-acoustic analysis of oceanic ambient noise. | Accepted | In this study, the dense seismo-acoustic network of the Institute of Geophysical Research (IGR), National Nuclear Centre of the Republic of Kazakhstan is used to characterize the global ocean ambient noise. As the monitoring facilities are co-located, this allows for a joint seismo-acoustic analysis of oceanic ambient noise. |
| 9 | Line 11 The abstract should be rebalanced more. It reads a bit as if there is little novelty in the findings, as the results that are stated have been reported on in previous studies.

Please try to bring out more what this study brings to literature. | Accepted, the abstract is rewriitten | |
| 10 | Line 14 The IGR network includes stations that are part of several national and global monitoring systems. | Accepted | Deleted |
| 11 | Line 14 not very relevant for an abstract; it is not essential to understand the study. | Accepted | Deleted |
| 12 | Line 16 – 17 The measurements are compared with microbarom and microseism source model output that is | Accepted | The measurements are compared with microbarom and microseism source model output that are |
| 13 | Line 18 range-independent | Accepted | Range-independent |
| 14 | Line 18 Surely for microseisms you cannot use this relation but you will have to use a different relation. Can you specify? | Accepted, corrected | The attenuation of microseisms is calculated taking into account seismic attenuation and bathymetry effect. |
| 15 | Line 24 As discussed in my original review, it appears to me that there is no evidence for this at all in this manuscript. The authors will have to prove this, based on cross-bearing of array processing results and/or inclusion of | The statement is proved, based on cross-bearing of array processing results and inclusion of microbarom simulations < (Figure C2 in Appendix C). | |

| | | | |
|---|---|---|---|
| | microseism/microbarom simulations.  It could be equally well be that the microbarom/microseism sources are in the patch between 0-20 N latitude.  If detections cannot be associated to possible source regions on the southern hemisphere, every statement related to the southern hemisphere in this manuscript should be removed. | | |
| 16 | Line 25 I miss some discussion of the microseism results in the abstract and I miss in the abstract what this joint study has brought. So what do we learn from joint observations? What is surprising?   And why is it important to characterize microseisms and microbaroms? | Accepted, corrected | These results reveal the strengths and weaknesses of seismic and acoustic methods and lead to the conclusion that a fusion of two techniques brought the investigation to a new level of findings. Summarized findings are also perspective for a better description of the source (localization, intensity, spectral distribution) and bonding mechanisms of the ocean/atmosphere/land interfaces. |
| 17 | Line 28 The introduction is still a bit confusing as you often swap from microseisms to microbaroms. This makes it complicated to follow for the reader who is not familiar with infrasound and/or microbaroms. Don't forget that this journal is "Solid Earth", so a lot of seismologists are reading this too.  I would recommend to start with microseisms as these were observed and described first in literature. First observations, followed by microseism theory and modeling.   After that, switch to infrasound and introduce that for a paragraph. Just state the basics of what infrasound is. Finally, introduce microbaroms as the atmospheric counterpart of microseisms and review observational, theoretical, and modeling studies. | Accepted | The introduction is rewritten in accordance with the suggested plan |
| 18 | Line 29 seismic hum and microseisms can not be classified as the same one, according to literautre. seismic hum is generated by infragravity waves, but microseisms are generated by ocean gravity waves.   please the introduction to introduce these different phenomena correctly. | Accepted, deleted while introduction reduction | |

| 19 | Line 31 The introduction of microseism / microbarom models needs work. Longuet-Higgins never developed the microbarom source model, but worked on a theory for microseisms. Eric Posmentier started developing a theory of microbaroms based on this in 1967. Other scientists that worked on a microbarom source model was first developed by Brekshokvkikh (1973) and later extended by Waxler and Gilbert (2006), Waxler (2007) and more recently de Carlo (2019?) | Accepted, corrected | The primary microseism peak (around 0.07 Hz) is generated when ocean waves reach shallow water near the coast and interact with the sloping seafloor (Hasselmann, 1963). The secondary peak of microseisms (between 0.1 and 0.2 Hz) is generated by the interaction of ocean waves of similar frequencies travelling in opposite directions (Longuet-Higgins, 1950). Longuet-Higgins' theory explains how counter propagating ocean waves can generate propagating acoustic waves and create secondary microseisms by exciting the sea floor. Hasselmann (1963, 1966) generalized Longuet-Higgins' theory to random waves by investigating non-linear forcing of acoustic waves.

Posmentier (1967) started developing a theory of microbaroms based on the Longuet-Higgins' theory. A microbarom source model was first developed by Brekhovskikh (1973), later extended by Waxler and Gilbert (2006), Waxler (2007), and more recently by de Carlo (2020). |
| 20 | Line 31 His name was Longuet-Higgins. Change use throughout manuscript. | Accepted | Longuet-Higgins |
| 21 | Line 39 reword: accurately simulated | Accepted, deleted | Deleted |
| 22 | Line 39 Longuet-Higgins' | Accepted, deleted | Deleted |
| 23 | Line 50 end | Accepted, deleted after section redaction | Deleted |
| 24 | Line 50 Nuclear-Test-Ban | Accepted, deleted after section redaction | Deleted |
| 25 | Line 50 There is a significant difference here between microseisms and microbaroms. While propagation paths for microseisms can be either along the earth surface (Rayleigh waves) or through the Earth as bulk waves, all microbarom observations are typically along propagation paths that have undergone multiple bounces on the Earth surface. This difference also makes that you can backproject microseisms along a seismic ray (https://agupubs.onlinelibrary.wiley.com/doi/10.1029/2008GL036111). As in this paper both microbaroms and microseisms are | Accepted, added | There is a significant difference between microseisms and microbaroms. While propagation paths for microseisms can be either along the earth's surface as Rayleigh waves, or through the Earth as body waves (Gerstoft et al., 2008), microbarom observations are typically along propagation paths that have undergone multiple bounces on the Earth surface. |

| | | | |
|---|---|---|---|
| | compared, it is important to point out commonalities and differences. | | |
| 26 | Line 55 used for nuclear verification monitoring. | Deleted after section redaction | Deleted |
| 27 | Line 55 you still need to define this band. | Deleted after section redaction | Deleted |
| 28 | Line 57 This is not a logical argument as it is written now and really needs to be broken down as it does not make sense to a reader that is specialized in microseisms but has never heard about microbaroms and infrasound. I would explain this in a different section, after the introduction where you link microbaroms to the state of the (middle) atmosphere. In such a section, explain what controls the detection capability. Part of it is the propagation efficiency from source(s) to receiver and noise conditions that are local to the receiver. Then you can explain that to estimate the source intensity, one must have an estimate of the transfer function that describes the loss of energy along the propagation path. An essential input for the computation of this transfer function are atmospheric specifications of wind and temperature throughout the WHOLE atmosphere. Finally, as propagation in the stratospheric waveguide is most efficient over long ranges and will likely control your detections, the specifications in the middle atmosphere play a dominant role in explaining the microbarom observations. | Accepted, deleted after section redaction | |
| 29 | Line 60 Pieter studied more events, but now it is written as they all occurred in the three month dataset. Rewrite. | Accepted | Smets et al. (2014) compared microbarom observations with the expected values to study the life cycle of Sudden Stratospheric Warming events. |
| 30 | Line 64 The considered dense seismo-acoustic Kazakhstani network is operated by the Institute of Geophysical Research (IGR) of the National Nuclear Center of the Republic of Kazakhstan and includes both seismic and infrasound arrays." | Accepted | The considered dense seismo-acoustic Kazakhstani network is operated by the Institute of Geophysical Research (IGR) of the National Nuclear Center of the Republic of Kazakhstan and includes both seismic and infrasound arrays |
| 31 | Line 64 Can you indicate what these authors found? | Accepted | A first-order agreement between the observed and modelled trends of microbarom back-azimuth was shown. |

| | | | It was shown that infrasound measurements can provide additional integrated information about the structure of the stratosphere where data coverage is sparse. |
|---|---|---|---|
| 32 | Line 66 Reword this part to bring out the added value of your paper. "In this paper, we develop a synergetic approach to better constrain microbarom source regions and evaluate propagation effects. To this end, we apply the method developed by Hupe et al. (2018) to the dense Kazakhstani seismo-acoustic array network. | Accepted | In this paper, we develop a synergetic approach to better constrain microbarom source regions and evaluate propagation effects. To this end, we apply the method developed by Hupe et al. (2018) to the dense Kazakhstani seismo-acoustic network. The considered network is operated by the Institute of Geophysical Research (IGR) of the National Nuclear Centre of the Republic of Kazakhstan. It includes both seismic and infrasound arrays. |
| 33 | Line 70 reword, is not proper English. | Accepted | Since the pioneering work of Donn and Naini (1973), to our knowledge, this study is the first multi-year comparisons between observed and modelled ambient noise at co-located seismo-acoustic arrays. |
| 34 | Line 73 Sentence is incomplete. Comparisons between microbarom predictions and microseisms can not be possible right? | Accepted | In the last part, comparisons between the observed and modelled microbaroms and microseism are discussed. |
| 35 | Line 76 explain why more = better. | Accepted, explained | The signal correlation in such a dense network is significantly higher compared to sparser networks like the IMS. |
| 36 | Line 76 I suggest the following subsections to discuss the networks, 1.1.1. Infrasound array network .. 1.1.2 Seismic array network ... | Accepted | 1.1.1. Infrasound array network. .. 1.1.2 Seismic array network ... |
| 37 | Line 79 Should be capitalized if it refers to a region in a country. | Accepted | North-West |
| 38 | Line 81 the village of Makanchi | Accepted | the village of Makanchi |
| 39 | Line 82 What dataloggers are used? What is the sampling rate? Wind noise filters? | Accepted, information added | All arrays are equipped with a 24-bit digitizer with a sampling frequency of 20 Hz at IS31 and KURIS, and 40 Hz at MKIAR. Data logger parameters are listed in Table A1 (Appendix A). All stations are equipped with 96 port wind noise reducing system with pipe rosettes, except L1, L2, L3, and L4 elements at IS31 which are connected to 144 inlet ports (Marty, 2019). |
| 40 | Line 83 This is clear now. Please omit. | Acce[pted | Deleted |
| 41 | Line 84 Suggest rewording: By associating infrasound observables over the network, | Accepted | By associating infrasound observables over the network, both natural and anthropogenic |

| | both natural and antropogenic infrasound sources can be detected and characterized | | infrasound sources can be detected and characterized |
|----|----|----|----|
| 42 | Line 86 suggest separate subsection, 1.1.2 seismic array network. Also discuss the datalogger and sampling rate here. | Accepted | All the arrays equipped with 24-bit digitizers, the sampling frequency is 40 Hz everywhere. |
| 43 | Line 86 that are part... | Accepted | That are part |
| 44 | Line 86 as well as the | Accepted | as well as |
| 45 | Line 86 KKAR arrays which are part.. | Accepted | arrays which are part |
| 46 | Line 94 Discuss here the implications of the response of the sensors of the ABKAR, BVAR, KKAR, MKAR arrays that are now on line 88-89. | Accepted | The frequency response of the sensors at MKAR, ABKAR, and KKAR is not flat in the 0.1-0.3 Hz band; however, as the response information is given, one can correct for the drop in amplitude; the phase shift difference between instruments part of the same array is assumed negligible. |
| 47 | Line 94 Use same frequency axis as Figure 2. Include phase spectrum. | Accepted | Information is added at Appendix A |
| 48 | Line 103 This is weird. Is it just an outlier? For this reason it would be good to look at a spectrogram over a month to see if this is typical or not. | Accepted, deleted after processing of the longer spectrogram | |
| 49 | Line 101 This statement you cannot make from just one day. It would be useful to include spectrograms to show this over the timeframe of month. | Accepted, checked via processing of the longer spectrogram, rephrased | The microbarom peak is more pronounced in October and December. |
| 50 | Line 106 why not 1 band if the settings are the same anyway? | Standard configuration is used Ceranna, R. Matoza, P. Hupe, A. Le Pichon, M. Landès Systematic array processing of a decade of global ims IMS infrasound data Infrasound Monitoring for Atmospheric Studies, Springer (2019), pp. 471-482, 10.1007/978-3-319-75140-5_13 | |
| 51 | Line 107 again, why not just 1 band from 0.1 to 0.4 Hz? | Standard configuration is used | |
| 52 | Line 116 This is indeed a function of signal SNR (for example as quantified through the least-squares error that would be estimated). How is that taken into account in this study? I understand that it is estimated that the SNR is taken constant, but what value is chosen for the calculation of the | Accepted, the value is represented | Uncertainties in wave parameter estimates are calculated considering the array geometry of the above mentioned infrasound and seismic arrays, assuming perfectly coherent signals and time delay errors bounded by twice the sampling period (Szuberla and Olson, 2004) (Table 1). |

| | | | |
|---|---|---|---|
| | error ellipses? It is important to state all assumptions, so that the results and calculations can be replicated. | | |
| 53 | Line 118 This section needs some restructuring and rewording as it is quite convoluted and therefore hard to understand. | Accepted | Reworded in accordance with the reviewer comments. |
| 54 | Line 119 Rephrase: In this research, we utilise the microseism source model output that is .... | Deleted after section redaction | Deleted |
| 55 | Line 119 and is calculated | Deleted after section redaction | Deleted |
| 56 | Lin 121 Add: Throughout this paper, this model output is referred to as 'p21'. | Deleted after section redaction | Deleted |
| 57 | Line 125 already stated | Deleted after section redaction | Deleted |
| 58 | Line 126 double comma | Deleted after section redaction | Deleted |
| 59 | Line 126 rephrase long-range microbarom propagation is determined by... | Deleted after section redaction | Deleted |
| 60 | Line 127 Reword this part as it is unclear since you are convoluting microbarom source modeling and propagation here. Of course microbarom modeling is not dependent on the dynamical properties of the middle atmosphere. With microbarom and microseism modeling you describe the source and that is it. So: (1) discuss the source model characteristics and then (2) discuss what determines the propagation structure from source to receiver). | Accepted | Part is rewritten |
| 61 | Line 132 It is important here to state that the De Carlo model suggests that microbarom source model is essentially a scaled version of the Hasselmann integral. Perhaps you can first introduce the Hasselmann integral which you can use to discuss the non-linear ocean-wave interaction. The Hasselmann integral also comes back in the discussion of the microseism source model. | Accepted, the Hasselmann integral is introduced | |
| 62 | Line 137 This should be separate from the source modeling because it is a propagation model. | Accepted, the item is separated | |
| 63 | Line 143 Explain how the number of predicted arrivals is computed. This is | Accepted, explained | The correlation is evaluated for the back-azimuths and amplitudes. |

| | | | |
|---|---|---|---|
| | unclear. How do changing background noise conditions play a role? | | |
| | Line 147 Better to show the global model so we have a better idea what the actual global distribution of microbarom sources is. As for example shown in Ouden et al., 2020 it can be quite complex. This data should already be there as the microbaroms are computed for the global grid (line 134). In that map you could then indicate with a white contour what you interpret to be the expected sources are that are observed (e.g. by applying cross-bearing localization). | Accepted | Figure C1 shows the averaged distribution of the expected microbarom amplitude over the globe. The calculation was carried out for two summer months. White iso-contours map the density of the microbarom source distribution. The sources were located via cross-bearing for the following station pairs: IS31-KURIS, IS31-MKIAR and KURIS-MKIAR. |
| 64 | Line 147 epicenter is not appropriate here, certainly in a geophysics context where it typically relates to earthquake, | The part is deleted | |
| 65 | Line 159 What does F_p represent? The Hasselmann integral? | Deleted after section redaction | |
| 66 | Line 159 what is K? | Deleted after section redaction | |
| 67 | Line 162 is the bottom compressible or the water? clarify | The ocean is compressible, this is the reason why we have seismic waves (P waves or pressure waves) propagating in the ocean. | |
| 68 | Line 172 Already stated. Remove. | Accepted, deleted | |
| 69 | Line 179 This observation is strange as it suggests that the observations would be BELOW the low noise model. Is there a calculation error somewhere? I would also expect the higher end of the amplitude range (0.1 Pa) to be somewhat higher. | Accepted, entire microbarom and microseism datasets were recalculated. Initially, the amplitude plots were built with the RMS amplitudes. In some cases, PMCC produces zero amplitudes that lower the average. Exchange on to peak-to-peak value improved the measurement accuracy. | ~ 0.005 – 0.5 Pa |
| 70 | Line 178 Is the frequency content of the summer detections different from the winter detections? It would be good that as it can give important information on the microbarom sources. | Accepted, the frequencies were measured; the average values at Summer are lower that means that epicentral distances are bigger | During the summer months, signals with back-azimuths of 210±50° dominate with a period ranging from 4 to 6.5 s and lower amplitude (~0.01 Pa) (Figure C1), suggesting waves propagating over longer epicentral distances. |
| 71 | Line 184 Also here I think the amplitudes are off. Should be at least higher than 0.001 Pa. You can also see that there is a large bias between model and observation. | Accepted, deleted | |

| 72 | Line 186 I can't really see this cluster and suspect a little that they are based on the modeling results. Overall, it seems that the summer detections for all arrays are rather scattered between, say, 45-225 degrees. | Accepted, deleted. | |
|---|---|---|---|
| 73 | Line 196 You should deconvolve with the sensor response so a fair comparison can be made. | Accepted. As it is not realistic to apply deconvolution to the waveforms and to recalculate PMCC detections, amplitude correction has been applied to the amplitudes of detections. The value corresponding to the detections with maximum amplitudes are chosen with a corresponding dominant frequency close to 0.2 Hz. Amplitude damping for this frequency was corrected for the instrument responses. Also, sensitivity issues were corrected for the MKAR and Kurchatov Cross. | |
| 74 | Line 209 For infrasound, the summer detections don't seem to be as well predicted as the winter detections. In particular, the model predicts more than what is detected. Is this because PMCC cannot detect multiple sources and is limited to the nearest source (in the south?) Also, it appears that during the Jan-Feb 2017 the simulations suggest that there is a SSW, however the observations show something else. Can you discuss that? | Accepted, the issue is discussed | Overall, at all stations, there is good agreement between the predicted and observed amplitudes during the winter months (Figure 2 d,h-l), but in summer, the predicted amplitudes are overestimated (Table 2). A first reason is that PMCC cannot detect multiple sources in the same frequency band. A second reason is the limitation of the propagation modelling which considers range independent atmosphere. It can be noted that the propagation anomaly predicted during of the SSW on January-February 2017 is not observed. Wind noise variations at the station, not considered in the simulations, could explain part of these discrepancies. |
| 75 | Line 209 I would agree for the infrasound data that there is good agreement, but there are significant differences for the microseism analysis. The microseism predictions show significant source regions to the south of the arrays that are not observed. Can you discuss why there is this discrepancy? | Accepted, discussed in the Discussion section | To summarize, both amplitudes and azimuths of the microbaroms are well predicted in winter as opposed to summer months. Microseism predictions show dominant source regions south of the arrays that are not observed. Quantitative estimations of the prediction quality (Scorr calculated according to Eqs. 3 and 4) are summarized in Table 2. |
| 76 | Line 223 This should be reconsidered as the reported amplitudes are likely incorrect. | It was reconsidered. After reprocessing the amplitudes are higher than NLNM. | |

| | | | |
|---|---|---|---|
| 77 | Line 226 I would agree for the infrasound data but there are significant differences for the microseism analysis. Can you comment? | Accepted. Commented in the discussion section | • The range of back-azimuths for North Atlantic microseisms is larger than the ones of microbaroms at ABKAR and MKAR as shown by Figure 5 (a,b,e,g). In winter, at ABKAR, signals with back-azimuth of ~310° are predicted, while the observed signals dominate at ~340°. In summer, the signals predicted around ~180° are not observed (Figure 3 (a)). Such deviations in surface wave back-azimuths were earlier identified during teleseismic events observation at Alp Array (Kolinsky, 2019). To substantiate this hypothesis, Source Specific Static Corrections (SSSC) are required. However, the SSSC evaluation would require long-term instrumental observations and in aseismic regions, which is out of the scope of the present studies. |
| 78 | Line 246 Can you include this evidence for this from the array processing? | The explanation is changed | These peaks are explained by North Pacific microbaroms. |
| 79 | Line 251 Figure 8 (IS31) shows that the predictions are much more diffuse and are scattered over the 45-225 degree back azimuth? | Accepted, the note is added | although there is a large spreading in the predictions (45-225°). |
| 80 | Line 251 MKIAR? | Accepted | MKIAR |
| 81 | Line 256 A more... | Accepted | A more |
| 82 | Line 256 There is no evidence for this. You need to include localization results to prove this. | Evidence is presented in the Appendix C | |
| 83 | Line 267 Indeed it is clearly shown in the simulations but not very in the observations. I see very few orange dots for IS31 and KURIS. Can you discuss why this is? Could it be that it was a minor SSW that allowed for bi-directional ducting conditions? These conditions have been seen before during a SSW (Assink et al., 2014). In the study by Ouden et al, 2020 we have shown that during such conditions, multiple microbarom sources can be detected that appear as concurrent sources in the records. The array processing method that is used in this study cannot accomodate concurrent sources and may likely only show the dominant (North Atlantic) source. | The reason is more related to the low signal to noise ratio for microbaroms at windy sites | |

| 84 | Line 297 It is not clear if microbaroms/microseisms from the southern hemisphere are detected, as discussed before. | The evidence is presented in Appendix C | |
|---|---|---|---|
| 85 | Line 302 not shown yet. needs crossbearing localisation analysis. | Evidence is in the Appendix C | |
| 86 | Line 305 I think this is overselling the result a bit. Correlation coefficients of 0.5-0.6 is not very high. Generally it is considered to be moderate, | Accepted | moderate |
| 87 | Line 307 That may be, but that is a bit odd to state in the conclusions.  It should be discussed (but in more detail) in the discussion. | Accepted and detailed and moved to the Discussion section | The results show that exploiting the synergy between seismic and infrasound ambient noise observations is valuable to: (i) better constrain the source strength using seismic records as microseisms propagate through the static structure of the Earth, while microbaroms travel through a highly variable atmosphere both in space in time, (ii) improve the detectability of ocean-wave interaction and location accuracy as microbarom wave parameters are less affected by heterogeneities in the propagation medium, and (iii) improve the physical description of seismo-acoustic energy partitioning at the ocean-atmosphere interface. |
| 88 | Line 311 I would also add that characterization of the noise field around the arrays is important for succesful verification of the CTBT using the IMS. | Accepted, added | Finally, including additional data from other seismo-acoustic networks worldwide would help constraining microbarom source location, validating long-range propagation modelling, and better characterize station-specific ambient noise signatures, which is important for a successful verification of the CTBT using the IMS. |
| 89 | Line 317 I think this statement would have been good a bit earlier on in the paper. | Accepted, moved to the Discussion | Additional studies are also required to further evaluate whether the bathymetry effect could explain discrepancies between the observed microbarom and microseism signals (Longuet- |

| | | | Higgins, 1950; Stutzmann et al., 2012, De Carlo, 2020). |
|---|---|---|---|
| 90 | Line 460 The | Accepted | The |
| 91 | Line 459 Describe which sites (Kurchatov (Kurchatov Cross/KURIS) and Makanchi (MKAR/MKIAR)) | Accepted | Seismic and infrasound arrays are collocated at Kurchatov (Kurchatov Cross/KURIS) and Makanchi (MKAR/MKIAR)). |
| 92 | KKAR and MKAR seismic arrays | Accepted | KKAR and MKAR seismic arrays |
| 93 | Line 460 The callouts | Accepted | The inset graphs |
| 94 | Line 462 Please include information on the datalogger in this plot as well. As the sampling rate is likely lower, the response curve is more likely to drop above 10 Hz. At least discuss using words. Ideally, this information would also be part of the figure. | The information on the dataloggers is listed in the Table A1, Appendix A | |
| 95 | Line 462 Discuss how you computed the vertical scale. The dB relative to what? The sensitivity of the Chaparral? Does this mean that the nominal sensitivity of the MB2000/MB2005 is 20 dB less than the Chaparral M25? | Accepted, corrected

Bode Gain Plots display the ratio of the system gain at each frequency. The magnitude of the transfer function T is defined as:

$$T(j\omega) = \sqrt{R^2 + X^2}$$

The frequency response is defined as:

$$T(j\omega) = \frac{\prod_n |jw + z_n|}{\prod_n |jw + p_n|}$$

Calculating the decibel yields:

$$Gain = \sum_n 20\log(|j\omega + z_n|) - \sum_m 20\log(|j\omega + p_m|)$$

https://en.wikibooks.org/wiki/Control_Systems/Bode_Plots | |
| 96 | Line 462 Can you include a phase spectrum too? | The phase spectrum is included | |
| 97 | Line 464 Use same frequency axis as Figure 2. Include phase spectrum. | The same frequency axis is used, phase spectrum is included. | |

| | | | |
|---|---|---|---|
| 98 | Line 468 Y-axis is off. Should be something like [dB re 20e-6 Pa**2/Hz]. And not "displacement" | Accepted, Y axis is corrected | PSD (dB) re Pa^2/Hz, the response units are listed in the table A1 |
| 99 | Line 468 Can you also include a figure where you show the seismic and infrasonic spectra in one figure for the co-located stations? This could be important for your findings. | Accepted | The figures for KURIS/Kurchatov Cross are included |
| 100 | Line 468 I recommend using a logarithmic x-axis because then you may also see the primary microseism band | Accepted, the log x-axis is used | |
| 101 | Line 474 Better to show the global model so we have a better idea what the actual global distribution of microbarom sources is. As for example shown in Ouden et al., 2020 it can be quite complex. In that map you could then indicate with a white contour what you interpret to be the expected sources are that are observed (e.g. by applying cross-bearing localization). | Accepted, the global model is shown in Appendix C | |
| 102 | Line 478 Better to show the global model so we have a better idea what the actual global distribution of microseism sources is. As for example shown in Ouden et al., 2020 it can be quite complex. In that map you could then indicate with a white contour what you interpret to be the expected sources are that are observed (e.g. by applying cross-bearing localization). | Accepted, the global model is shown in Appendix C | |
| 103 | Line 478 Use the same map boundaries as Figure 6. It would also be very interesting to show side-by-side maps for microbaroms and microseisms for the timeperiods. | The picture is omitted | |
| 104 | Line 484 It appears that in Jan-Feb 2017 there was a SSW that clearly shows in the simulations but not in the observations. Can you comment? Is this a limitation of PMCC? | The lack of detection is likely due to strong wind noise. | |
| 105 | Line 484 This observation is strange as it suggests that the observations would be BELOW the low noise model (Brown et al., 2010). There must be a calculation error somewhere? | Accepted, the amplitudes are corrected. The peak-to-peak amplitudes are used instead of RMS ones. | |
| 106 | Line 489 Also here the amplitude is (likely) off. | Accepted, the amplitudes are corrected. The peak-to-peak | |

| | | | amplitudes are used instead of RMS ones. |
|---|---|---|---|
| 107 | Line 494 Can you explain the discrepancy between the source predictions around ~180 degrees and the observations? | The explanation is given in the discussion section | The range of back-azimuths for North Atlantic microseisms is larger than the ones of microbaroms for ABKAR and MKAR (Figure 5 (a), (b), (e), and (g)). Observed values differ from simulation data significantly at all stations. E.g., for ABKAR for winder period, signals with back-azimuth ~310° were predicted, however the actual back-azimuths of observed signals were ~340°. Deviation of simulated vs. observed back-azimuths is the greatest at this array. In summer months, instead of expected signals with back-azimuths 180°, signals with back-azimuths 290° were registered. There is no evident common rule in deviation behaviour. Such huge deviations in surface wave back-azimuths were identified earlier during teleseismic events observation at Alp Array (Kolinsky, 2019). To substantiate this hypothesis, Source Specific Static Corrections (SSSC) are required. It should be mentioned however that the SSSC evaluation requires long-term instrumental observations, and in aseismic regions it can take decades. (Here and below) |
| 108 | Line 497 Also here is a discrepancy around ~180 degrees. | The explanation is given in the discussion section | |
| 109 | Line 499 Again, a discrepancy around 180 degrees. | The explanation is given in the discussion section | |
| 110 | Line 499 Can you discuss this bias around 300 degrees? Could it be related to the station-specific slowness-azimuth corrections that should be applied? | The explanation is given in the discussion section | |
| 111 | There are large differences between the microseism predictions and observations. Could this be due to the station specific slowness-azimuth corrections that should be applied? For example https://pubs.geoscienceworld.org/ssa/bssa/article/89/4/989/102851 | Yes. In our opinion, the SSSC is the most probable reason for the discrepancy. We made attempt to obtain the SSSC's for the North Atlantic region. The trends of correction are right, but its absolute values do not fully cover the discrepancy. These results were presented at GA EGU and later develop in the PhD thesisof Smirnov. Authors SSSC evaluation experience | |

| | | shows that ~20 year period of the Kazakh network operation is not sufficient to collect enough GT events. | |
|---|---|---|---|
| 112 | Line 503 Here there is a slightly better match around ~180 degrees. | The explanation is given in the discussion section | |
| 113 | Line 503 There is a 'line' of detections around 120 degrees that is not predicted. Any idea? The band towards the NE-E is also not detected. Can this be explained? | For the 120 degrees we have no idea. The NE-E maybe connected to icequakes, e. g. (Mikhailova, Komarov., 2009). https://www.nnc.kz/media/bulletin/fil es/G3yHUIJY0n.pdf The study is scheduled at KazNDC for the nearest future to resolve the nature of these sources. | |
| | In some small amount of cases, the reviewer's suggestion to make insignificant language corrections are accepted without being mentioned in this list. | | |

[revised manuscript text omitted]
 **6-hour consecutive time windows.**

[Figure]

**Figure D1.** Azimuthal distribution of infrasound detections throughout 2017 (a), from December 1, 2016, to February 28, 2017 (b), and from June 1 to August 31, 2017 (c). Azimuthal distribution of seismic detections throughout 2017 (d), from December 1, 2016, to February 28 (e), 2017, and from June 1 to August 31, 2017 (f).

---

## Author Response (AR3)

Dear Jelle,

Thank you for your appreciation of the paper in its current iteration, it is undoubtedly tied to the substantial contribution from your side.
We have no objection to your comment regarding the assumed location of microbaroms generation registered by the Kazakh network in the summer.
We do not refuse our hypothesis, however, we completely agree with you that its wording is rather bold, considering the available evidence data is limited. We have 'toned down' all affected points and noted that additional surveys are required to collect sufficient proof or negative the hypothesis. Possible ways of obtaining incorrect interpretation results are specified individually.
This subject was discussed in the paper in five (5) places, all items and amendments are summarised in the table below.

Kind regards
Alexandr Smirnov

| # | Comments from Referees #2 | Author's response | Author's changes in manuscript |
|---|---|---|---|
| 1 | Line 22 … while signals observed in summer likely originate from sources located in the southern hemisphere. | Statement modified | ...while signals observed in summer could originate from sources located in the southern hemisphere, however additional analyses are required to consolidate this hypothesis. |
| 2 | Line 191 Figure C2 shows the averaged global source distribution of microbarom sources in summer. The sources are located via cross-bearing considering detections at IS31, KURIS and MKIAR. A hotspot is located southwest of South America. | The sentence and explanation are moved to the discussion section | |
| 3 | Line 210. The distances to the source regions differ essentially from summer to winter. For example, simulations predict three source regions at IS31 in winter. Distances to the two regions in the North Atlantic are around 3500 km and 7000 km, and about 7000 km to the North Pacific. In summer, one source region is located in the Pacific Ocean, and two other sources in Southern high latitudes at | | The distances to the source regions differ essentially from summer to winter. For example, simulations predict three source regions at IS31 in winter. Distances to the two regions in the North Atlantic are around 3500 km and 7000 km, and about 7000 km to the North Pacific. In summer, one source region is located in the Pacific Ocean and two other sources at Southern high latitudes at distances of ~12000 km and ~18000 km. However, the calculation of attenuation using a range-independent atmospheric model would inevitably lead to great mistakes in such situation. |

| | | | |
|---|---|---|---|
| | distances of ~12000 km and ~18000 km. | | |
| 4 | Line 254 (Discussions) A more complex picture is observed in summer. Some stations detect signals from regions along the peri-Antarctic belt while simulations predict microbaroms with larger amplitude. Other stations detect signals southward, but the detected back-azimuths disagree with the predictions. | | Microbarom sources recorded by the Kazakh network in summer are not fully characterized. The cross-bearing location considering detections at IS31, KURIS, and MKIAR yields a hotspot located southwest of South America (Figure C2). Since the localization does not include crosswind effect, the true location may differ significantly from the preliminary estimation. Furthermore, the fact that a signal should pass a considerable portion of the way upwind, would prejudice the likelihood of its registration. However, this preliminary location is consistent with the relatively low amplitude values and larger periods in summer than in winter (Figure C1). Additional studies using more realistic propagation modelling are required to confirm this hypothesis. |
| 5 | Line 316 (Conclusions) In summer, microbarom detections at IS31, KURIS and MKIAR are consistent with ocean storms located along the peri-antarctic belt southwest of South America, at distances larger than 15000 km from the arrays, which is consistent with the relatively low amplitude and frequency of the recorded signals. | | In summer, the location of microbarom signals using detections at IS31, KURIS, and MKIAR is found southwest of South America, at a distance larger than 15000 km, near the peri-Antarctic belt where strong ocean storms circulate. This location is consistent with the relatively low amplitude and frequency of the recorded signals. |

---

## Author Response (AR4)

Dear Charlotte,

Although there are no changes were requested some corrections were done. The content (e.g. text, figures, equations, or tables) of all files uploaded in this form is exactly the same as the version of my manuscript accepted by the Topical Editor. However, the resolution of the figures is enlarged to 300 dpi and the legends are added to figures 2, 3, and C2. The text and manuscript files contain the last version of the figures. Also, some format failures were found and corrected.

Kind regards

Alexandr Smirnov